# Landscape Conservation Forecasting for Data-Poor at-Risk Species on Western Public Lands, United States

**Louis Provencher** [1,*], **Kevin Badik** [1], **Tanya Anderson** [2], **Joel Tuhy** [3], **Dan Fletcher** [4], **Elaine York** [5] and **Sarah Byer** [1]

1   The Nature Conservancy, 1 East First Street, Reno, NV 89501, USA; kevin.badik@tnc.org (K.B.);
    sarah.byer@tnc.org (S.B.)
2   The Nature Conservancy, Las Vegas, NV 89101, USA; tanderson@tnc.org
3   The Nature Conservancy, Moab, UT 84532, USA; jtuhy@tnc.org
4   U.S. Department of Interior, Bureau of Land Management, Cedar City, UT 84720, USA; dfletche@blm.gov
5   The Nature Conservancy, Salt Lake City, UT 84102, USA; eyork@tnc.org
*   Correspondence: lprovencher@tnc.org; Tel.: +1-775-322-4990 (ext. 3190)

**Abstract:** Managing vast federal public lands governed by multiple land use policies creates challenges when demographic data on at-risk species are lacking. The U.S. Bureau of Land Management Cedar City Field Office used this project in the Black Mountains (Utah) to inform vegetation management supporting at-risk greater sage-grouse and Utah prairie dog planning. Ecological systems were mapped from satellite remote sensing imagery and used to model species habitat suitability under two levels of management activity (custodial, preferred) and climate scenarios for historic and two global circulation models. Spatial state-and-transition models of ecological systems were simulated for all six scenarios up to 60 years while coupled with expert-developed habitat suitability indices. All ecological systems are at least moderately departed from reference conditions in 2012, whereas habitat suitability was 50.5% and 48.4% for sage-grouse and prairie dog, respectively. Management actions replaced non-native annual grasslands with perennial grasses, removed conifers, and controlled exotic forbs. The drier climate most affected ecological departure and prairie dog habitat suitability at 30 years only. Different climates influenced spatial patterns of sage-grouse habitat suitability, but nonspatial values were unchanged. Climate impacts on fire, vegetation succession, and restoration explain many results. Front-loading restoration is predicted to benefit under future drier climate.

**Keywords:** greater sage-grouse; Utah prairie dog; state-and-transition spatial simulation models; ecological departure; climate scenarios; southwest Utah





## 1. Introduction

Local public land managers face challenges when planning conservation of sensitive, threatened, or endangered (hereafter at-risk) species in multiuse lands. Managers must assign high priority to the management of ecological systems supporting at-risk species while considering other natural resource and economic values [1]. Furthermore, managers have limited funds to achieve all management goals.

Local demographic data help guide management actions to improve at-risk species habitat, but it is uncommon to have complete demographic data on at-risk species for more than one drought cycle (e.g., seven-year El Niño cycle) reflecting highs and lows of reproductive effort in most western states—it is even uncommon for the simplest demographic data [2]. Where demographic, behavior, and landscape data are available, habitat suitability and per capita population growth rates can be statistically estimated [3,4].

Public land managers preparing plans to conserve whole multiple-use landscapes may be concerned with the following questions [5]: (1) What is the current condition of vegetation? (2) What is the current condition of the habitat for at-risk species? (3) Is vegetation likely to get worse over time under status quo (hereafter, custodial) land management? (4) Will at-risk species habitat decline under custodial land management? (5) What kind

of management actions, how much per year, and at what cost will meaningfully improve altered systems and habitat over a period of time? (6) Which actions produce the highest ecological return-on-investment given limited funding?

Advances in scenario-based state-and-transition simulation models (STSM) methods and software feasibly allow land managers working with modelers to address these questions that straddle ecological system management planning and single species management [5–9], but seldom incorporate climate change scenarios [10]. STSMs are computer simulations of parameterized box-and-arrows models where each box represents a vegetation class (i.e., states) of an ecological system [8]. The Nature Conservancy's Landscape Conservation Forecasting™ (LCF) method combines vegetation layers obtained from remote sensing with STSMs to compare the effects of alternative management or climate scenarios on vegetation condition and other metrics [5,9,10]. Inspired by the Landscape Fire and Resource Management Planning Tools project (known as LANDFIRE, www.landfire.gov, accessed on 14 January 2012, [11]), past LCF projects only focused on the ecological departure of vegetation—the dissimilarity of each ecological system's vegetation classes between the forecasted scenario and reference or desired condition [11]. Ecological departure can guide site planning management because departure can be partitioned to different vegetation classes [5,9]. However, vegetation responses to management do not always translate to change in species' habitat suitability because ecological departure is nonspatially estimated by system, whereas habitat suitability is spatially estimated across entire landscapes. Using habitat suitability as a metric of condition to guide vegetation treatment implementation would increase the likelihood of success for the targeted species.

The main impetus of this project was to incorporate habitat suitability as an additional metric of condition into Landscape Conservation Forecasting for two at-risk species managed by the Bureau of Land Management (BLM) in southwestern Utah: greater sage-grouse (*Centrocercus urophasianus*; hereafter, GSG) and the federally threatened Utah prairie dog (*Cynomys parvidens*; hereafter, UPD). Both species lacked local, long-term research to inform traditional demography in this region. BLM was concerned about two future climate change risks: (a) climate warming would affect the habitat of GSG and increase the likelihood of extirpation because this is the most southern mid-elevation population of the species and (b) future two- and three-year droughts, respectively, preceding and following the year of restoration actions including seeding would cause seedings to fail and become dominated by non-native annual species. Objectives to support the BLM's management include (a) building spatially explicit habitat suitability indices for GSG and UPD, (b) comparing the information from a nonspatial ecological departure metric to habitat suitability, (c) comparing the effects of additional active versus custodial management on ecological systems and habitat suitability, and (d) determining the amount of compensatory vegetation management required to mitigate habitat degradation under future climate scenarios.

## 2. Materials and Methods

Three parts comprised our methods: remote sensing of vegetation data, STSM modeling, and metric development. We will provide a summary as our methods have evolved over time and been detailed elsewhere [5,6,9,10]. STSM reviews and examples can be found in Czembor and Vesk [7] and Rumpff et al. [12] for uncertainty analysis in older nonspatial STSM, Provencher et al. [10] for application to management, uncertainty accounting, and climate change, and Daniel et al. [8] for theoretical and software advancements in nonspatial and spatial STSM.

### 2.1. Study Area

The Black Mountains are in the southeastern part of the Great Basin ecoregion of southwest Utah (126,053 ha centered about the coordinates 38°01′52.17″ N, 113°02′49.87″ W; Figure 1). Monsoonal storms are more frequent in this landscape than in the rest of the Great Basin to the west [13]. The Black Mountains are a circular low mountain range

surrounded by extensive subxeric flat desert to the south and west, and by the rapid rise of the Utah High Plateau to the east. The geology is primarily volcanic. Vegetation is zonal with subxeric shrublands at lower elevations, woodlands at middle and higher elevations, and sagebrush shrublands at middle to higher elevations. The low salt desert scrub valley floor is at 1580 m elevation. The GSG population is small and one of the most southern in the species' range [14]. The local UPD population is one of the healthiest in its current range [15].

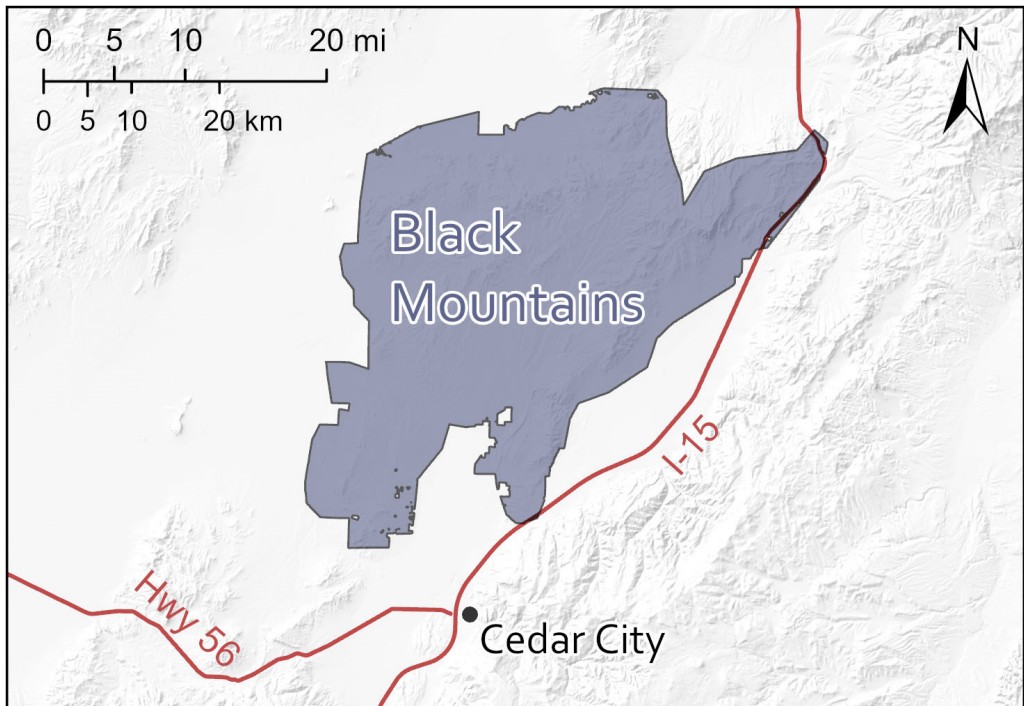

**Figure 1.** Black Mountains project area in southwestern Utah north of Cedar City, UT, USA. Project area approximate coordinates are 38°01′52.17″ N, 113°02′49.87″ W. Legend: I-15 is Interstate Highway 15 and HWY 56 is Highway 56.

### 2.2. At-Risk Species

GSG historic range has declined by 56% in 11 of 13 historically occupied western U.S. states, and two of three historically occupied Canadian provinces in landscapes dominated by sagebrush shrublands [14]. About 71% of GSG habitat in the U.S. Fish and Wildlife Service (USFWS) southern Great Basin management zone applicable to this study is under the jurisdiction of the Bureau of Land Management [14]. Other than lek counts, most BLM Field Offices have not collected local or regional GSG movement or demographic data. GSG is a sagebrush (*Artemisia* spp.) obligate species that requires large tracts of continuous sagebrush cover dotted with wet meadows and nonforested riparian corridors. Age diversity of sagebrush patches and moist vegetation provide resources for the distinct breeding, summer, and winter seasonal habitats of GSG. In the southern Great Basin portion of the species' range the USFWS has identified uncharacteristically large wildfires, lack of policies protecting habitat, increased fuels from invasive annual grasses, and conifer encroachment into shrublands as contributors to habitat loss and the greatest threats to GSG [14].

UPD is currently found in the southwest quarter of Utah in a fraction of its original range [15]. The species' decline since the 1920s was caused by poisoning, non-native plague, fire exclusion, overgrazing, land use change, and invasive species [15]. The species is found from 1646 to 2896 m elevation. UPD lives in shrublands but does not tolerate shrubs taller than 0.5 m and more than 15% shrub cover [15,16]. Unlike other prairie dogs, it does not

actively remove woody vegetation surrounding its burrows. It is speculated that UPDs occupied the early successional phase of shrublands after fires [17].

### 2.3. Remote Sensing of Vegetation Layers

STSMs and metric estimation required two vegetation layers from remote sensing: ecological systems and their vegetation classes. Ecological systems (also known as biophysical settings [11]) are potential vegetation types expected in the physical environment under natural disturbance regimes usually named for the dominant upper-layer vegetation (e.g., Wyoming big sagebrush). Vegetation class is the ecological system's current status defined by canopy structure, succession, and whether it is in reference or uncharacteristic condition. The nested description of 27 potential systems and up to 23 classes per system resulted in >250 unique combinations (Supplemental Table S1) that were modeled and potentially mapped by remote sensing.

Remote sensing was conducted from RapidEye 5 m resolution multispectral satellite imagery from 19 June 2013. Moreover, freely available 1 m resolution NAIP imagery and Google Earth imagery (www.earth.google.com/web/) were used to assist 5 m imagery with small tree detection. Following an unsupervised classification of imagery using the software Imagine® from Leica Geosystems, field surveys were conducted from 30 September to 5 October 2012, and 20 June to 2 July 2013 to verify systems and classes. More than 6000 rapid observation points were collected to ensure that a large percentage of the landscape was visited. At each rapid observation location, the ecological system, vegetation class, explanatory notes, and at least two georeferenced photographs were taken. In addition to these observations, spatial data of seeding treatments that followed fires but predated our mapping were used to improve the interpretation of imagery. Local agency experts reviewed remote-sensed maps for final needed corrections.

### 2.4. State-and-Transition Simulation Modeling

We resampled the system-class rasters from their original 5 m resolution to 50 m due to computational limits imposed by hardware for STSM. To avoid losing small, ecologically important vegetation types, ecological systems and vegetation classes were resampled according to a user-defined hierarchy (Supplemental File S2). Small or linear ecological systems and vegetation classes, and systems critical to species success were given higher priority than common systems, which were resampled with majority rule. These final rasters represent "current condition" vegetation.

STSMs were simulated for 60 years using ST-Sim, a module in the SyncroSim platform (www.ApexRMS.com; www.syncrosim.com, both accessed on 14 January 2020, [8]). ST-Sim simulations share many characteristics with Markov chains. Added components, such as management actions, area implementation targets, and time dependent functions, distinguish ST-Sim simulations from Markov chains [8]. In ST-Sim, each pixel was assigned an initial condition state (a state is the combination of an ecological system, usually static, and a vegetation class) obtained from remote sensing that can either (a) age one time step and stay in the same class, (b) age one time step into an older class (i.e., succession), or (c) experience a probabilistic disturbance and transition to ≥1 other states, including the originating state. Transitions are probabilistic (ecological disturbances and, sometimes, succession) or deterministic (succession to another class after a fixed number of years). Land management actions were implemented using area targets (e.g., 1000 ha·yr$^{-1}$ seeded on average in designated vegetation classes). Probabilistic disturbances and management actions can be modified or constrained temporally or spatially to mimic real world processes such as climate variability, fire spread behavior, and equipment operation limits.

Fire disturbances had widespread influence on vegetation structure and management actions. The mean fire return interval per ecological system was calculated based on (1) nonspatial reference simulations without post-European influences that were run for 700 years to reach equilibrium and (2) the duration of succession classes, each with different fire return intervals (Table 1). Presettlement fire return intervals used by reference

simulations were suppressed by 90% for management simulations because current fire management precludes most natural fire regimes. The record of unique fires from the federal fire occurrence data [18] for 1980 to 2016 and Monitoring Trends in Burn Severity data [19] from 1984 to 2016 showed that the BLM's exclusion success was about 90% both in terms of number of fires and area burned, compared to reference simulation results.

**Table 1.** Reference mean fire return interval (MFRI) of ecological systems assuming presettlement vegetation and dynamics. The MFRI is the mean weighted by the area of each class in the ecological system.

| Ecological System | MFRI (Year) |
| --- | --- |
| Aspen Woodland | 118 |
| Aspen–Mixed Conifer | 64 |
| Basin Wildrye | 46 |
| Big Sagebrush semi-desert | 126 |
| Black Sagebrush | 141 |
| Curl-leaf Mountain Mahogany | 283 |
| Desert Wash | 5447 |
| Four-Wing Saltbush | 1541 |
| Greasewood–Basin Big Sagebrush | No fire |
| Juniper Savanna | 276 |
| Limber–Bristlecone Pine | 350 |
| Low Sagebrush | 252 |
| Mixed Conifer | 95 |
| Mixed Salt Desert | No fire |
| Montane Riparian | 159 |
| Montane Sagebrush Steppe | 51 |
| Pinyon–Juniper | 262 |
| Ponderosa Pine | 20 |
| Semi-Desert Grassland | 101 |
| Stansbury Cliffrose | 131 |
| Utah Serviceberry | 45 |
| Wet Meadow–Montane | 67 |
| Winterfat | 1258 |
| Wyoming Big Sagebrush upland | 113 |

## 2.5. Spatial Modeling

Species' habitat suitability required the use of spatial STSMs to account for distances of critical landscape features. Spatial modeling is more complex than nonspatial modeling because additional data are needed. Initial conditions in the STSMs included rasters of ecological systems, vegetation classes, and land ownership (Supplemental File S3).

### 2.5.1. Constraints on Management Actions

Rasters that spatially constrain implementation of management actions were uploaded to ST-Sim (Supplemental File S3). Based on the experience of BLM machinery operators, all actions that used tractors pulling seeders, mowers, or harrows were limited to slopes less than 15%. Masticators, chainsaws, and chains dragged by two bulldozers could be used on slopes up to 30%.

The simulations also required two at-risk species spatial constraints. First, all actions that thinned sagebrush were prevented within 50 m of leks. Second, two aerial or ground-based treatments unique to UPD that combined aerial seeding, aerial herbicide spraying to reduce woody vegetation (e.g., Tebuthiuron), and mechanical soil disturbance (harrow and chaining) focused on greatly reducing woody cover were implemented primarily inside (99% of cases) and marginally outside (1% of cases) designated UPD management areas (Figure 2).

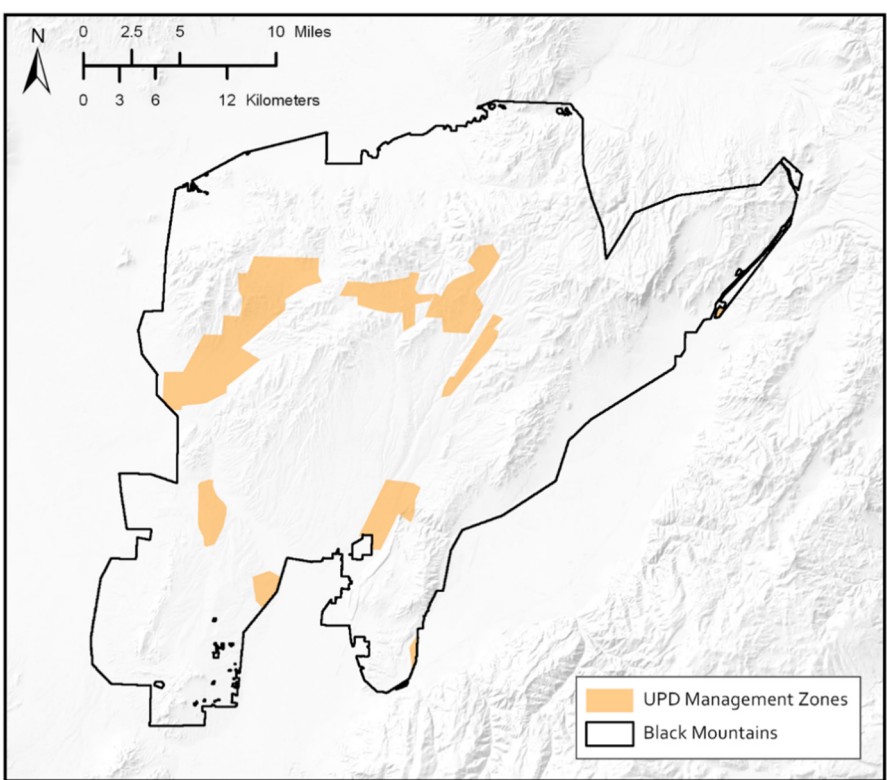

**Figure 2.** UPD priority management zones in the Black Mountains. The probability of implementing two vegetation thinning treatments unique to UPD is 1.0 in the management zones and 0.1 outside of them.

2.5.2. Fire

Spatial disturbances were not all mapped. The size and spread of fire activity also determined where management actions were applied in the simulations (Tables 2 and 3). Unique fires data from [18,19] were used to calculate fire size distribution in the Black Mountains. Fire spread in the Black Mountains was modeled using three principles: (1) prevailing winds elongate fire predominantly from the southwest to the northeast, while allowing other directions (Figure 3); (2) fire spreads more rapidly upslope than downslope relative to wind directions based on McArthur's fire danger meter ([20], Table 3); and (3) natural fire ignition locations are spatially determined by observed lightning strikes and anthropogenic ignition locations near roads. Lightning strike locations obtained from the BLM were converted to a frequency map using a trial-and-error 1.5 km (0.94 mile, or $39 \times 39$ pixels) moving window. The frequency values were standardized from 0 to 1 and converted into a 50 m resolution raster of lightning strike density to model natural fire starts. Pixels with values of 1 had the highest likelihood of fire starts via lightning. A second raster of human-caused ignitions was modeled using the distance from frequently used roads. Based on Morrison's [21] ignition data, distances from roads were also standardized to values between 0 and 1 using the equation:

$$H(i) = 1.0171 \cdot exp[-0.004 \times Dist(i)]$$

where *H(i)* = probability of human ignition at pixel *i* and *Dist(i)* = distance from pixel *i* to the nearest road with frequent use. The maximum value between the two layers for each pixel was retained to create the final map of ignition likelihood. Once fire ignited in these locations, the fire spread based on underlying vegetation characteristics and prevailing wind directions.

**Table 2.** Size distribution (ha) of fire events for the Black Mountains (UT) based on federal fire occurrence data from 1980 to 2016 and the Monitoring Trends in Burn Severity (MTBS) data from 1984 to 2016. For example, a size class of "≤4" indicates fire events were ≤4 ha.

| Area of Disturbance (ha) | Black Mountains Percent Occurrence |
|---|---|
| ≤0.10 | 64 |
| >0.10 to 4 | 20 |
| >4 to 40 | 7 |
| >40 to 121 | 2 |
| >121 to 405 | 4 |
| >405 to 2024 | 2 |
| >2024 to 8097 | 1 |

**Table 3.** Fire spread slope multipliers using McArthur's fire danger meter [20].

| Slope (%) | Multiplier |
|---|---|
| −16 | 0.4700 |
| −8 | 1.1320 |
| 0 | 1.0000 |
| 8 | 0.5600 |
| 16 | 3.9620 |

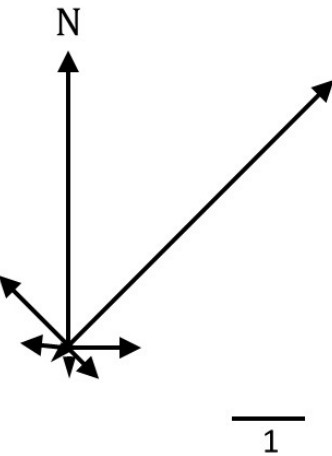

**Figure 3.** Fire spread direction multiplier imitating prevailing winds. Multipliers preferentially spread fire pixel to pixel. N is north.

*2.6. Management Scenarios*

A management scenario is a group of land management actions and specific climate effects that define a simulation theme. Six management scenarios were simulated for 60 years. The six scenarios combine two land management levels with three future climates.

CUSTODIAL MANAGEMENT represents only baseline management actions of fire suppression and livestock management. PREFERRED MANAGEMENT includes active management in addition to baseline management actions. While USD 0 was assigned to the CUSTODIAL MANAGEMENT scenario, the PREFERRED MANAGEMENT scenario was limited to maximum annual expenditures levels of USD 1 million from years 1 to 10 with an emphasis on lower elevation sagebrush systems, USD 1 million from years 11 to 25 with

an emphasis on higher elevation sagebrush systems, USD 500,000 from years 26 to 45, and USD 100,000 from year 46 to 60 for maintenance activities. If simulated treatments cannot find enough areas to treat, realized expenditures will be less than the maximum allowed. A list of management actions was selected by agency experts per ecological systems. Each action was assigned a cost per area (Table 4) and other implementation attributes were imbedded in the simulation library: success and failure proportions, vegetation class outcomes for success and failures, and slope constraints.

**Table 4.** Cost of management actions per unit area (USD/ha). Note: Plateau® is the commercial name for the herbicide imazapic that inhibits seed germination of annual plant species.

| Ecological System/Action | Area Cost (USD/ha) |
|---|---|
| Aspen Woodland | |
| Fence | 24,700 |
| Black Sagebrush | |
| 2 × Chaining + Plateau + Seed | 543 |
| 2 × Chaining + Seed | 469 |
| Chaining + Plateau + Seed | 358 |
| Chaining + Seed | 296 |
| Herbicide-Plateau + Seed | 272 |
| Masticate + Herbicide + Seed | 766 |
| Masticate + Native-Seed | 803 |
| Masticate + Seed | 667 |
| Small-Tree-Lopping | 198 |
| Thin | 86 |
| Low Sagebrush | |
| Aerial-Spike + Aerial-Native-Seed + Chain-Harrow | 865 |
| Chaining + Native-Seed | 432 |
| Spot-Herbicide + Native-Seed | 556 |
| Montane Riparian | |
| Chainsaw-Thinning | 198 |
| Exotic-Control | 889 |
| Fence | 19,760 |
| Weed-Inventory + Treat | 889 |
| Montane Sagebrush Steppe | |
| Aerial-Spike + Aerial-Seed + Chain-Harrow | 494 |
| Chaining + Plateau + Seed | 457 |
| Herbicide-Plateau + Seed | 272 |
| Masticate + Native-Seed | 803 |
| RxFire + Seed + Chain | 580 |
| Pinyon–Juniper | |
| Chainsaw-Thinning | 198 |
| Stansbury Cliffrose | |
| Spot-Herbicide + Native-Seed | 556 |

**Table 4.** *Cont.*

| Ecological System/Action | Area Cost (USD/ha) |
| --- | --- |
| Utah Serviceberry | |
| Chainsaw-Thinning | 568 |
| Herbicide-Plateau + Seed | 371 |
| Masticate + Native-Seed | 803 |
| Wet Meadow–Montane | |
| Exotic-Control | 889 |
| Fence | 19,760 |
| Weed-Inventory + Treat | 790 |
| Wyoming Big Sagebrush upland | |
| Aerial-Spike + Aerial-Native-Seed + Chain-Harrow | 865 |
| Aerial-Spike + Aerial-Seed + Chain-Harrow | 494 |
| Chaining + Native-Seed | 432 |
| Chaining + Plateau + Seed | 457 |
| Chaining + Seed | 296 |
| Herbicide-Plateau + Seed | 272 |
| RxFire + Seed + Chain | 580 |
| Small-Tree-Lopping | 148 |
| Thin | 86 |

Future climates were combined with the management levels. The "control" climate scenario was historic climate data from 1950 to 2018 obtained from the Parameter–elevation Relationships on Independent Slopes Model (PRISM; [22]). This period captured the severe droughts of the 1950s and represented better climate station instrumentation after the Second World War. For future climate warming, two LOcalized Constructed Analogs (LOCA; Western Regional Climate Center's SCENIC site, [23,24]) were selected by local stakeholders to compare a worst-case future climate to the observed climate trend. The ACCESS1 LOCA represents the worst-case climate with the driest conditions in all seasons. The CCSM4 LOCA was selected by BLM managers and TNC staff because its projections reflect their observations about increasingly wetter spring and summer seasons, and drier and hotter winters.

The ACCESS 1.0 is one of two versions of the Australian Community Climate and Earth System Simulator coupled model (ACCESS-CM) that was built by coupling the UK Met Office atmospheric Unified Model UM, and other submodels, to the ACCESS Ocean Model (ACCESS-OM), a coupled ocean–sea ice model consisting of the NOAA/GFDL ocean model MOM4p1 and the LANL sea ice model CICE4.1, under the CERFACS OASIS3.2-5 coupling framework [25]. The Community Climate System Model (CCSM) is a coupled global climate model (GCM) developed by the University Corporation for Atmospheric Research (UCAR) [26]. CCSM4 includes four submodels (land, sea ice, ocean, and atmosphere) connected by a coupler that exchanges information with the submodels.

Minimum and maximum temperature, and precipitation time series were annually averaged across the landscape to capture the temporal variability for each climate. The six management–climate scenarios that combined with precipitation and minimum and maximum temperature yielded 18 (=6 scenarios × 3 climate time series) time series. In addition to temperature and precipitation, the 100-year time series of future $CO_2$ from the A2 scenario (aggressive business-as-usual emissions scenario) was downloaded from IPCC [27]. A stochastic weather generator (SWG) [28] was used to replicate each climate time series 10 times over 65 years. Estimating severe drought in any year, which considers

climate conditions for the previous five years [29], required that the time series be simulated over 65 years rather than 60 years with the first of the five years starting in year 1945. The $CO_2$ time series was not replicated as it has no variability and smoothly trends upward over time.

The purpose of simulating future climate was to introduce temporal variability in dominant ecological processes [10]. Variability directly affects processes through temperature and precipitation, or indirectly mediates processes through the Standard Precipitation Index (SPI). SPI is a standardized drought index based on precipitation and is expressed in positive (wet) and negative (dry) standard deviations from the mean [30,31]. SPI was calculated using the "spei" function in R package 'SPEI' [32]. The time series created with the SWG were the inputted climate data to the "spei" R function. To establish a climate warming difference before and after mapping (year 2013), the R code replicated future time series after 2013 while statistically considering the historic period from 1945 to 2013.

Temperature, precipitation, or SPI time series were transformed into transition multipliers in ST-Sim [10]. Transition multipliers are a quantitative means to determine how climate variations influence ecological processes. A transition multiplier is a varying annual unitless number $\geq 0$ in an annual time series that multiplies a base disturbance rate in the STSM. For example, a transition multiplier of one implies no change in the annual probability for fire, a transition of zero is a complete suppression of fire, and a transition of three triples the annual probability of fire.

A transition multiplier can be obtained from empirical data or theoretically derived. Transition multipliers are also the mechanism by which replicates are created for each scenario. Transition multipliers are determined by dividing each yearly value of the time series (for example, area burned) by the temporal average of the time series, thus creating a nondimensional time series with an average of one, which means that the whole temporal multiplier time series has no effect on the model's base rate. The process of associating time series of temperature, precipitation, SPI, and $CO_2$ with specific ecological processes deserves fundamental research (Supplemental File S4) [10]. Ecological processes whose variability was affected by climate were: Exotic Species Invasion, May Hard Freeze (aspen woodland only), 36 month drought mortality, 24 month drought mortality, Severe Drought, Shrubland Fire Activity, Shrubland and Forest Fire Activity, Wet Year, Very Wet Year, Annual Species Invasion, Tree (native) Invasion, and Flooding (Supplemental File S4).

### 2.7. Range Shifts

Climate change included simulated range shifts, which are the replacement of "cooler or wetter" ecological systems and their indicator species by "warmer or drier" systems and their indicator species due to climate warming. Localized range shifts are rarely studied in the literature. While complex, range shifts only contribute a small amount to area dynamics; therefore, the description of methods is found in Supplemental File S5 (also see Provencher et al. [10]).

### 2.8. Unified Ecological Departure

Unified ecological departure is the first of three metrics of condition. Traditional ecological departure was pioneered by the LANDFIRE program and is the dissimilarity between the observed distribution of vegetation class percentages and the predicted presettlement or natural distribution of vegetation class percentages obtained from presettlement equilibrium simulations. The latter is called the natural range of variability, per ecological system (*NRV*; [5]):

$$Ecological\ Departure\ (ED) = 100\% - \sum_{i=1}^{R} \min\{Observed\%(i),\ NVR\%(i)\}$$

where *i* is the number of *R* reference classes.

Unified ecological departure is coded into ST-Sim's menu *Ecological Departure* and begins with ecological departure calculated as above, and then (1) scores the departure higher (makes condition worse) according to levels of vegetation class undesirability present (e.g., noxious forbs) and (2) scores the departure slightly lower (makes condition slightly better) according to agreed-upon management threshold levels of allowable uncharacteristic classes present (e.g., introduced species seeding):

$$Unified\ Ecological\ Departure\ (UED) = Min\left(100,\ Max\left[\begin{array}{c} 0, ED - \sum_{i=R+1}^{U(No-Thresh)} \min\{HRF(i), Observed\%(i), 0\} - \\ \sum_{j=U(No-Tresh)+1}^{N} \min\{Threshold\%(j), Observed\%(j)\} \end{array}\right]\right)$$

where *R, U(No-Thresh),* and *N* are, respectively, number of reference classes, uncharacteristic classes without threshold values, and total vegetation classes. *Threshold(j)* is a user-supplied management threshold for class *j*, and *HRF(j)* is the high-risk function of class *j* for different levels of "undesirability." Uncharacteristic vegetation classes with an undesirability level >0 are assigned a high risk value based on the arbitrary function *HRF* selected based on desirable curve fitting properties. We chose a negative sigmoid function for *HRF*:

$$HRF(j) = -\exp(c(B-1))/(1+\exp(c(B-1)))$$

where c is an arbitrary fitted coefficient (here 10) and B is the undesirability level from the table. *HRF* = 0, −0.5, and −1 for, respectively, values of B = 0, 1, and 2. When thresholds and *HRFs* are not specified in ST-Sim, the *UED* equation simplifies to the *ED* equation.

*2.9. Greater Sage-Grouse Habitat Suitability*

Rasters of current and simulated vegetation were used to estimate current and future species habitat suitability. These metrics identify areas that would benefit most from management and restoration. Restoring already good GSG habitat wastes limited resources as these areas will remain good habitat without intervention. Poor habitat areas far from leks or wet systems would not benefit from restoration due to strong distance limitations. Coupling spatial STSMs and a custom R script enabled the model to place management actions in areas of intermediate GSG habitat suitability where efforts would be most beneficial.

GSG habitat suitability was estimated for each pixel with a custom-made R program used as a stand-alone application, or dynamically coupled to ST-Sim to constrain implementation of management actions. Overall GSG habitat suitability was the average of three groups of seasonal Resource Selection Functions (*RSF*; [4]). Seasonal habitats were nesting, summer, and winter. The *RSF*s described below were the result of a GSG expert workshop held on 13 March at TNC's office in Salt Lake City.

Heuristic *RSF*s were developed because there were only very limited movement data from collared GSG from the southern UT populations. Using GSG demographic and movement data from the entire State of Utah, experts assisted with defining the shape of resource selection functions that had the strongest effect on GSG habitat suitability. Resource selection functions were scaled from 0 (not suitable) to 1 (very suitable). The independent variables for the different resource selection functions were distance to the closest critical attribute (e.g., type of vegetation, busy road), the proportion of a resource in the surrounding habitat, or the value of a pixel's vegetation type as seasonal habitat.

Five resource selection functions (order of presentation is not related to importance and all *RSF*s were assumed equal) characterized the nesting season (i.e., nesting; *RSF(N,i)*; see Supplemental File S6 for detailed equations):

*RSF(N,1)*: Distance of each pixel (nest site) to the closest lek—habitat suitability was high and increased up to a maximum at 5 km from the closest lek and then rapidly decreased with distance.

*RSF(N,2)*: Distance of each pixel (nest site) to the closest trees—habitat suitability increased up to 2 km from trees, and farther than 2 km habitat was fully suitable.

*RSF(N,3)*: Proportion of pixels with adequate shrub cover 1000 m around each pixel (nest site)—habitat suitability increased with the proportion of adequate pixels.

*RSF(N,4)*: Distance of each pixel (nest site) to the closest busy road—habitat suitability increased with distance with the most severe reduction of suitability at less than 150 m from busy roads followed by a rapid increase, and no effect after 5 km.

*RSF(N,5)*: Resource selection function was equal to the expert-defined value of the vegetation class (Supplemental Table S6.1) to nesting habitat for each pixel (nest site).

Nesting habitat suitability index (*HSI[N]*) equals mean{*RSF(N,1)*, *RSF(N,2)*, *RSF(N,3)*, *RSF(N,4)*, *RSF(N,5)*}.

Four resource selection functions characterized the summer (i.e., brood-rearing) season (*RSF[S,i]*):

*RSF(S,1)*: Distance of each pixel to the closest high-elevation shrubland pixels above 2134 m elevation that are late-brood habitat or to the closest wet meadow—habitat suitability was high up to 3 km from each pixel and then rapidly decreased with distance reaching zero at 12 km.

*RSF(S,2)*: Distance of each pixel to the closest lek—habitat suitability remained high up to 5 km from each pixel to the closest lek and then decreased rapidly until it was nearly zero at 10 km.

*RSF(S,3)*: Distance of each pixel to the closest trees—habitat suitability increased up to 2 km from trees and after 2 km habitat was fully suitable (same function as for nesting).

*RSF(S41)*: Resource selection function was equal to the expert-defined value of the vegetation class (Supplemental Table S6) to summer habitat for each pixel.

Summer habitat suitability index (*HSI[S]*) equals mean{*RSF(S,1)*, *RSF(S,2)*, *RSF(S,3)*, *RSF(S,4)*}.

Three *RSF*s characterized the winter season (*RSF[W,i]*):

*RSF(W,1)*: Distance of each focal (bird's location) pixel to a distant pixel, which is only considered if the proportion of adequate winter shrub pixels (values > 0.3 from Supplemental Table S6) in a 1015 m moving window from the distant pixel is >75%—if the distant pixel is acceptable, habitat suitability was 1 km up to 5 km from the closest lek and then rapidly decreased with distance.

*RSF(W,2)*: Distance of each pixel to the closest low or black sagebrush pixel—habitat suitability linearly decreased up to a distance of 25 km and then became zero.

*RSF(W,3)*: Resource selection function was equal to the expert-defined value of the vegetation class (Supplemental Table S6) to winter habitat for each pixel.

Winter habitat suitability index (*HSI[W]*) equals mean{*RSF(W,1)*, *RSF(W,2)*, *RSF(W,3)*}.

Finally, the overall habitat suitability (*HSI*) across all seasons was the average of seasonal habitat suitability multiplied by Simpson's evenness index:

$$\text{Overall habitat suitability} = 1/3 \times (HSI(N) + HSI(S) + HSI(W) \times \sum_{i=i}^{N,S,W} p(i)^2$$

where *p(i)* is *HSI(i)*/(*HSI*(N) + *HSI*(S) + *HSI*(W)). The resulting overall value per pixel was between 0 (not suitable) and 1 (maximally suitable). Statistical habitat suitability models are not constructed with multiple *HSI*s or an evenness index (e.g., [33,34]). However, in the absence of sufficient demographic data, the above methods account for both the contribution of seasonal habitat suitability and whether some seasonal habitats were deficient and, as a result, lowered the overall habitat suitability [33,34] as the evenness index does.

## 2.10. Utah Prairie Dog Habitat Suitability

The habitat suitability estimation for UPD was composed of only two *RSF*s as this species does not have seasonal life cycles and its limiting habitat preferences are simple. The

first *RSF* (*RSF (1)*) examined the proportion of acceptable forage vegetation (herbaceous) within 110 m of a pixel, the estimated foraging distance for UPD [15]. If a mix of pixels with unacceptable and acceptable vegetation occurred in the window, habitat suitability was calculated as the proportion of acceptable pixels. If the proportion of unacceptable, nonhabitat vegetation (e.g., true pinyon–juniper woodlands, aspen, riparian, curl-leaf mountain mahogany) exceeded 85%, the habitat suitability was zero.

The second *RSF* considered the distance to the closest prairie dog colony pixel using two linear equations:

$$RSF(2) = \begin{cases} 1, \text{ if distance to closest colony pixel } \leq 3 \text{ km} \\ 0.5 \times \text{distance to closest colony pixel } + 1, \text{ if } 3 \text{ km} < \text{distance} < 5 \text{ km} \\ 0, \text{if distance } \geq 5 \text{ km} \end{cases}$$

## 3. Results

### 3.1. Initial Conditions

Twenty-two ecological systems, sparsely vegetated areas, and water bodies were mapped in the Black Mountains (Table 5 and Figure 4). The most prevalent systems with a combined area of 16,000 hectares were intermediate-elevation shrublands and woodlands including black sagebrush, montane sagebrush steppe, Wyoming big sagebrush upland, and pinyon–juniper woodland. Among the smallest systems were those dependent on higher levels of soil moisture (montane riparian, wet meadow–montane, saline meadow, desert wash) and those at low elevations found mostly outside the project area (big sagebrush semi-desert, mixed salt desert scrub, semi-desert grassland). Seven systems were moderately departed (33% < UED ≤ 66%) and 14 were highly departed (UED > 66%) from reference conditions (Table 5).

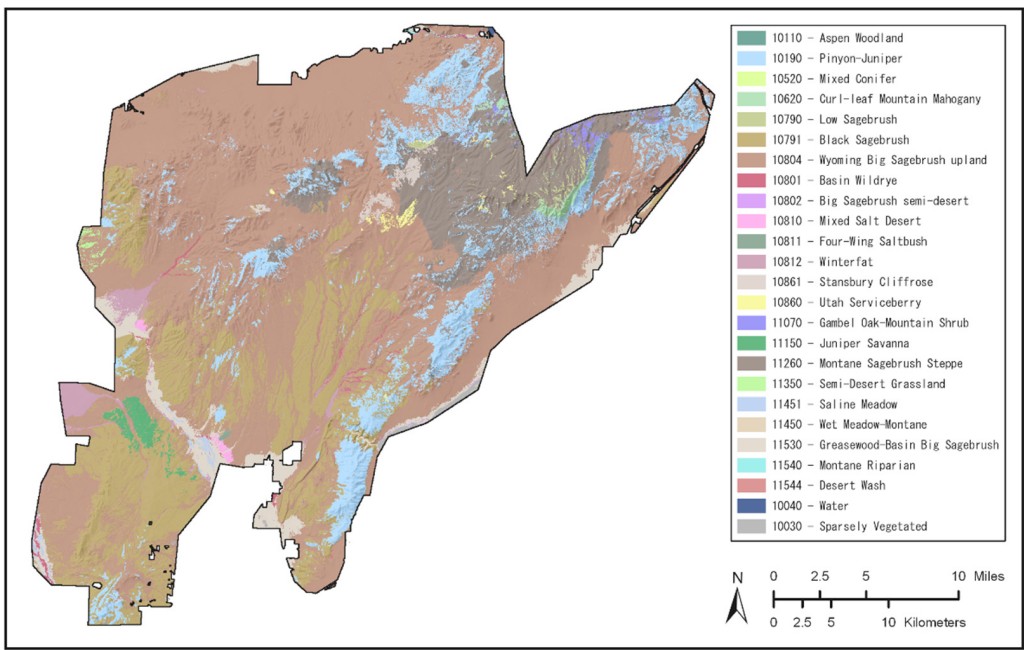

**Figure 4.** Ecological systems of the Black Mountains. Mapping was based on interpreted multispectral RapidEye 5 m resolution satellite imagery from 19 June 2013.

**Table 5.** Area (ha) and unified ecological departure (UED) of ecological systems mapped in the Black Mountains (UT). UED color legend: red is >66% and 33% < yellow ≤ 66%.

| Ecological System | Area (Hectares) | UED at Year 0 (%) |
|---|---|---|
| Wyoming Big Sagebrush upland | 85,971 | 99 |
| Black Sagebrush | 29,436 | 74 |
| Pinyon–Juniper | 16,806 | 100 |
| Montane Sagebrush Steppe | 16,690 | 40 |
| Greasewood-Basin Big Sagebrush | 5233 | 46 |
| Winterfat | 2004 | 100 |
| Juniper Savanna | 1120 | 36 |
| Stansbury Cliffrose | 834 | 42 |
| Basin Wildrye | 739 | 100 |
| Sparsely Vegetated | 637 | na |
| Curl-leaf Mountain Mahogany | 576 | 40 |
| Gambel Oak–Mountain Shrub | 554 | 89 |
| Low Sagebrush | 454 | 36 |
| Utah Serviceberry | 440 | 100 |
| Saline Meadow | 334 | 100 |
| Mixed Salt Desert | 232 | 100 |
| Montane Riparian | 231 | 90 |
| Semi-Desert Grassland | 200 | 63 |
| Wet Meadow–Montane | 76 | 93 |
| Four-Wing Saltbush | 52 | 66 |
| Water | 39 | na |
| Mixed Conifer | 35 | 97 |
| Big Sagebrush semi-desert | 32 | 100 |
| Aspen Woodland | 11 | 100 |

In 2012 (year 0), average GSG habitat suitability was 50.5% (Figure 5); average UPD habitat suitability was 48.4% (Figure 6). Although UPD population management zones (Figure 2) showed moderate to high suitability, the most suitable areas were to the north of the population management zones near the northern boundary.

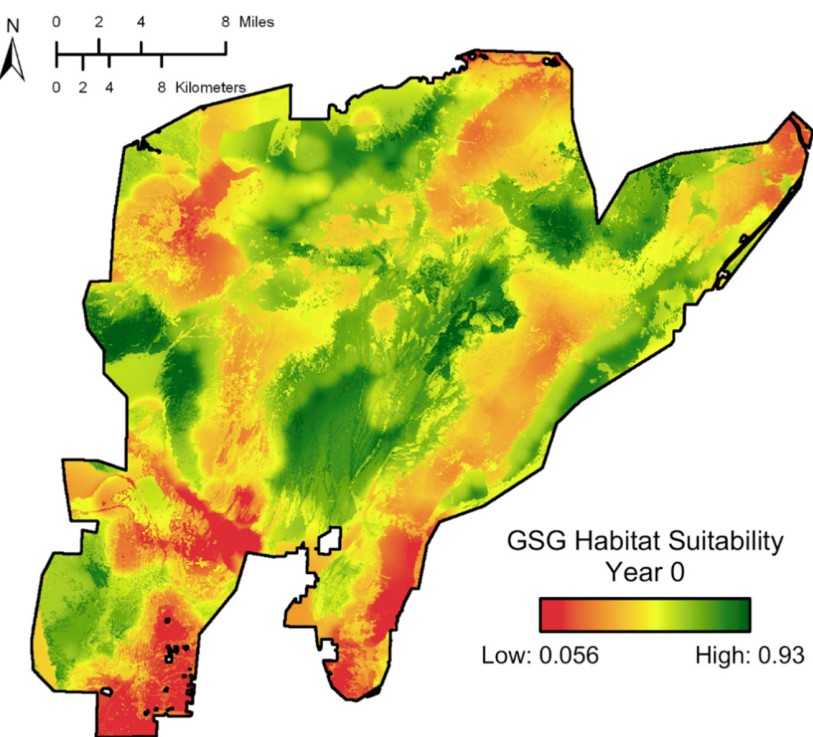

**Figure 5.** Current (year 0) habitat suitability index for GSG in the Black Mountains based on 2013 five-meter RapidEye satellite imagery. Average habitat suitability for year 0 of simulation was 50.5%. At year 0, there is no difference among management scenarios. Darker green indicates higher habitat suitability index, and redder indicates poorer habitat suitability index.

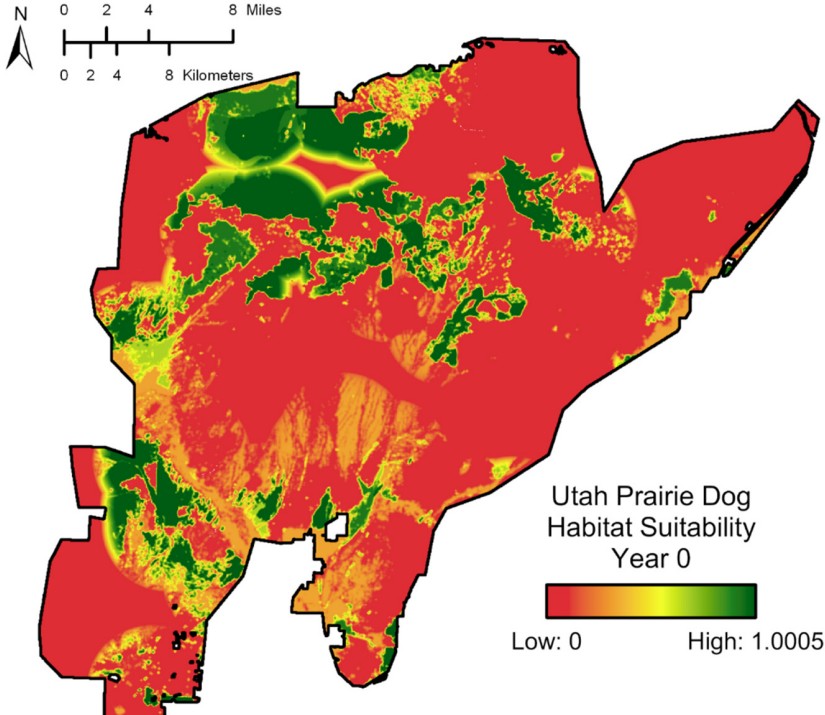

**Figure 6.** Current (year 0) habitat suitability index for UPD in the Black Mountains based on 2013 five-meter RapidEye satellite imagery. Average habitat suitability for year 0 of simulation was 48.4%. At year 0, there is no difference among management scenarios. Darker green indicates higher habitat suitability index, and redder indicates poorer habitat suitability index.

*3.2. Future Conditions with* CUSTODIAL MANAGEMENT

The CUSTODIAL MANAGEMENT scenario identified ecological systems that will further degrade, stay very highly degraded, or may need special attention for GSG, UPD, or weed (non-native annual species and exotic riparian forbs and trees) management. In addition, BLM staff requested that we include systems that are important to habitat of mule deer (*Odocoileus hemionus*) and migratory birds, although those species were not assessed here. These systems included aspen woodland (*Populus tremuloides*), black sagebrush (*Artemisia nova*), low sagebrush (*Artemisia arbuscula*), montane riparian, montane sagebrush steppe, Stansbury cliffrose (*Purshia stansburyana*), wet meadow–montane, and Wyoming big sagebrush (*Artemisia tridentata* spp. *wyomingensis*; see boldface type, Table 6). Utah serviceberry (*Amelanchier utahensis*) and pinyon-juniper woodland improved over time in CUSTODIAL MANAGEMENT scenarios due to fire that recruited older vegetation classes into under-represented and early successional classes (Table 6). Utah serviceberry was selected for active management because it is key browse for mule deer and is used by GSG. Minor management was modeled in pinyon–juniper woodland for fire risk management and pine nut harvesting, a cultural practice that stakeholders wanted BLM to protect. Thus, these 10 systems were chosen to receive special management actions in active management scenarios.

*3.3. Future Conditions with* PREFERRED MANAGEMENT

Two-way factorial ANOVA and preplanned contrasts were used to test the effects of management and climate scenarios. The preplanned contrasts compared the three climate scenarios of preferred management. At year 30, all ecological systems departed from reference condition (increased UED) except pinyon–juniper woodland and Utah serviceberry (Table 6). At year 60, all ecological systems moved closer to their reference conditions (decreased UED) except pinyon–juniper woodland and Stansbury cliffrose (Table 6).

Climate effects on UED were not detected in aspen woodland and wet meadow–montane at year 30, or for aspen woodland, black sagebrush, low sagebrush, Stansbury cliffrose, Utah serviceberry, and wet meadow–montane at year 60 (Table 6). UED was higher in the HISTORIC climate than in the LOCAs for black sagebrush at year 30, and in pinyon–juniper woodland at year 60. In systems where climate affected ecological departure, UED was lower at year 30 in the ACCESS1 LOCA than in the CCSM4 LOCA or historic climate for low sagebrush, montane sagebrush steppe, pinyon–juniper woodland, Stansbury cliffrose, and Utah serviceberry (Table 6).

After 30 years of PREFERRED MANAGEMENT, the cost was lower for the HISTORIC climate (USD 11.99 ± 0.62 95% C.I. million) than the ACCESS1 (USD 13.13 ± 0.35 million) or CCSM4 climate (USD 13.13 ± 0.89 million) (Table 6). The added total costs from years 26 to 60 for the HISTORIC, ACCESS1 and CCSM4 climates, respectively, were USD 8.89 ± 0.43 million, USD 6.29 ± 0.46 million, and USD 6.33 ± 0.43 million. Therefore, long-term costs were lower under climate change (Table 6).

**Table 6.** Mean unified ecological departure of ecological systems (% ± 95% C.I., n = 10) and cost of active scenarios 30 and 60 years later for all scenarios. Systems in boldface were selected for active management. The letters "a" and "b" adjacent to UED values denoted significantly different ($p \leq 0.05$) means based on the test of Custodial vs. Preferred management. The "i" and "ii" adjacent to UED values indicated two significantly different LOCA preplanned comparisons using contrasts: Historic vs. LOCAs and CCSM4 vs. ACCESS1. Montane riparian and Wyoming big sagebrush were not testable due to lack of variance across all management treatments. *UED* colors are explained in the caption of Table 5; moreover, the color green is *UED* $\leq 33\%$.

| Ecological System | CUSTODIAL + HISTORIC | CUSTODIAL + ACCESS1 | CUSTODIAL + CCSM4 | PREFERRED + HISTORIC | PREFERRED + ACCESS1 | PREFERRED + CCSM4 |
|---|---|---|---|---|---|---|
| | | | **Year 30** | | | |
| **Aspen Woodland** [&] | 98 ± 2 [a,i] | 99 ± 1 [a,i] | 99 ± 1 [a,i] | 76 ± 5 [b,i] | 76 ± 6 [b,i] | 78 ± 7 [b,i] |
| Basin Wildrye | 100 ± 0 | 100 ± 0 | 100 ± 0 | 100 ± 0 | 100 ± 0 | 100 ± 0 |
| Big Sagebrush semi-desert | 100 ± 0 | 100 ± 0 | 100 ± 0 | 100 ± 0 | 100 ± 0 | 100 ± 0 |
| **Black Sagebrush** | 96 ± 4 [a,i] | 85 ± 3 [a,ii] | 88 ± 5 [a,ii] | 84 ± 5 [b,i] | 76 ± 3 [b,ii] | 77 ± 4 [b,ii] |
| Curl-leaf Mountain Mahogany | 77 ± 7 | 44 ± 10 | 54 ± 13 | 67 ± 11 | 43 ± 11 | 60 ± 12 |
| Four-Wing Saltbush | 30 ± 17 | 42 ± 8 | 34 ± 15 | 40 ± 17 | 39 ± 8 | 27 ± 8 |
| Gambel Oak–Mountain Shrub | 62 ± 13 | 28 ± 10 | 52 ± 16 | 55 ± 14 | 33 ± 15 | 52 ± 15 |
| Greasewood–Basin Big Sagebrush | 100 ± 0 | 100 ± 0 | 100 ± 0 | 100 ± 0 | 100 ± 0 | 100 ± 0 |
| Juniper Savanna | 75 ± 12 | 31 ± 10 | 41 ± 17 | 65 ± 14 | 38 ± 14 | 55 ± 15 |
| **Low Sagebrush** | 77 ± 8 [a,i] | 41 ± 13 [a,ii] | 54 ± 18 [a,ii] | 58 ± 13 [b,i] | 28 ± 15 [b,ii] | 50 ± 14 [b,i] |
| Mixed Conifer | 41 ± 7 | 22 ± 5 | 26 ± 11 | 34 ± 12 | 25 ± 5 | 30 ± 12 |
| Mixed Salt Desert | 100 ± 1 | 100 ± 0 | 99 ± 1 | 100 ± 0 | 99 ± 1 | 99 ± 2 |
| **Montane Riparian** | 100 ± 0 | 100 ± 0 | 100 ± 0 | 100 ± 0 | 100 ± 0 | 100 ± 0 |
| **Montane Sagebrush Steppe** | 96 ± 3 [a,i] | 90 ± 3 [a,ii] | 93 ± 3 [a,i] | 82 ± 5 [b,i] | 72 ± 5 [b,ii] | 79 ± 5 [b,i] |
| **Pinyon–Juniper** | 73 ± 5 [a,i] | 42 ± 7 [a,ii] | 58 ± 13 [a,iii] | 64 ± 9 [a,i] | 43 ± 8 [a,ii] | 59 ± 10 [a,i] |
| Saline Meadow | 100 ± 0 | 100 ± 0 | 100 ± 0 | 100 ± 0 | 100 ± 0 | 100 ± 0 |
| Semi-Desert Grassland | 100 ± 0 | 100 ± 0 | 100 ± 0 | 100 ± 0 | 100 ± 0 | 100 ± 0 |
| **Stansbury Cliffrose** | 94 ± 5 [a,i] | 80 ± 8 [a,ii] | 84 ± 10 [a,ii] | 83 ± 8 [b,i] | 65 ± 10 [b,ii] | 81 ± 12 [b,i] |
| **Utah Serviceberry** | 78 ± 7 [a,i] | 57 ± 15 [a,ii] | 66 ± 13 [a,ii] | 67 ± 12 [a,i] | 40 ± 14 [a,ii] | 63 ± 11 [a,i] |
| **Wet Meadow–Montane** [&] | 85 ± 5 [a,i] | 92 ± 2 [a,i] | 89 ± 5 [a,i] | 40 ± 12 [b,i] | 41 ± 9 [b,i] | 42 ± 14 [b,i] |
| Winterfat | 100 ± 0 | 100 ± 0 | 100 ± 0 | 100 ± 0 | 100 ± 0 | 100 ± 0 |
| **Wyoming Big Sagebrush upland** | 100 ± 0 | 100 ± 0 | 100 ± 0 | 99 ± 1 | 94 ± 3 | 99 ± 2 |
| Total Cost (USD) 1–25 year (in $10^6$) | | | | 11.99 ± 0.62 | 13.13 ± 0.35 | 13.12 ± 0.89 |

**Table 6.** *Cont.*

| Ecological System | CUSTODIAL + HISTORIC | CUSTODIAL + ACCESS1 | CUSTODIAL + CCSM4 | PREFERRED + HISTORIC | PREFERRED + ACCESS1 | PREFERRED + CCSM4 |
|---|---|---|---|---|---|---|
| | | | Year 60 | | | |
| Aspen Woodland [&] | 95 ± 3 [a,i] | 95 ± 3 [a,i] | 98 ± 1 [a,i] | 87 ± 10 [b,i] | 83 ± 10 [b,i] | 80 ± 12 [b,i] |
| Basin Wildrye | 100 ± 0 | 100 ± 0 | 100 ± 0 | 100 ± 0 | 100 ± 0 | 100 ± 0 |
| Big Sagebrush semi-desert | 100 ± 0 | 100 ± 0 | 100 ± 0 | 100 ± 0 | 100 ± 0 | 100 ± 0 |
| Black Sagebrush | 99 ± 3 [a,i] | 95 ± 3 [a,i] | 97 ± 3 [a,i] | 79 ± 6 [b,i] | 81 ± 3 [b,i] | 75 ± 6 [b,i] |
| Curl-leaf Mountain Mahogany | 81 ± 4 | 72 ± 13 | 77 ± 6 | 79 ± 1 | 70 ± 11 | 71 ± 10 |
| Four-Wing Saltbush | 55 ± 16 | 67 ± 14 | 55 ± 10 | 29 ± 12 | 65 ± 8 | 56 ± 11 |
| Gambel Oak–Mountain Shrub | 53 ± 17 | 59 ± 14 | 58 ± 13 | 53 ± 12 | 46 ± 13 | 47 ± 15 |
| Greasewood-Basin Big Sagebrush | 100 ± 0 | 100 ± 0 | 100 ± 0 | 100 ± 0 | 100 ± 0 | 100 ± 0 |
| Juniper Savanna | 94 ± 8 [i] | 64 ± 12 [ii] | 76 ± 15 [ii] | 91 ± 5 | 80 ± 14 | 68 ± 14 |
| Low Sagebrush | 72 ± 10 [a,i] | 69 ± 12 [a,i] | 71 ± 15 [a,i] | 60 ± 8 [b,i] | 51 ± 11 [b,i] | 51 ± 19 [b,i] |
| Mixed Conifer | 58 ± 8 [i] | 42 ± 8 [ii] | 42 ± 13 [ii] | 49 ± 8 | 40 ± 6 | 36 ± 11 |
| Mixed Salt Desert | 100 ± 0 | 97 ± 4 | 96 ± 3 | 100 ± 0 | 91 ± 7 | 93 ± 6 |
| Montane Riparian | 100 ± 0 | 100 ± 0 | 100 ± 0 | 100 ± 0 | 100 ± 0 | 100 ± 0 |
| Montane Sagebrush Steppe | 98 ± 2 [a,i] | 95 ± 3 [a,ii] | 95 ± 3 [a,ii] | 78 ± 5 [b,i] | 72 ± 5 [b,ii] | 72 ± 5 [b,ii] |
| Pinyon–Juniper | 96 ± 6 [a,i] | 76 ± 14 [a,ii] | 82 ± 12 [a,ii] | 92 ± 7 [a,i] | 73 ± 11 [a,ii] | 76 ± 9 [a,ii] |
| Saline Meadow | 100 ± 0 | 100 ± 0 | 100 ± 0 | 100 ± 0 | 100 ± 0 | 100 ± 0 |
| Semi-Desert Grassland | 100 ± 0 | 100 ± 0 | 100 ± 0 | 100 ± 0 | 99 ± 1 | 100 ± 0 |
| Stansbury Cliffrose | 91 ± 7 [a,i] | 87 ± 8 [a,i] | 90 ± 8 [a,i] | 86 ± 10 [a,i] | 82 ± 11 [a,i] | 78 ± 9 [a,i] |
| Utah Serviceberry | 79 ± 7 [a,i] | 68 ± 15 [a,i] | 69 ± 9 [a,i] | 62 ± 11 [b,i] | 59 ± 11 [b,i] | 52 ± 13 [b,i] |
| Wet Meadow–Montane [&] | 99 ± 2 [a,i] | 99 ± 2 [a,i] | 99 ± 1 [a,i] | 49 ± 10 [b,i] | 47 ± 10 [b,i] | 57 ± 7 [b,i] |
| Winterfat | 100 ± 0 | 100 ± 0 | 100 ± 0 | 100 ± 0 | 100 ± 0 | 100 ± 0 |
| Wyoming Big Sagebrush upland | 100 ± 0 | 100 ± 0 | 100 ± 0 | 95 ± 4 | 97 ± 4 | 96 ± 3 |
| Total Cost (USD) 26–60 year (in $10^6$) | | | | 8.69 ± 0.43 | 6.29 ± 0.46 | 6.33 ± 0.46 |

[&] Required an arcsin(square-root(0.01 × X)) transformation to homogenize variances.

Because of BLM's keen interest in seeding success, it is noteworthy that different climates affected early-successional introduced grass seedings. During the first 25 years of intensive treatment implementation both PREFERRED + HISTORIC and PREFERRED + CCSM4 produced more seeded areas than the drier PREFERRED + ACCESS1 (Figures 7 and 8). After 30 years, the area of early-successional introduced species seeding remained high in the PREFERRED + HISTORIC scenario compared to PREFERRED + CCSM4 and PREFERRED + ACCESS1, which improves UPD habitat and diminishes GSG habitat. PREFERRED + CCSM4 and PREFERRED + ACCESS1 favored greater area of mid-successional introduced species seeding after 30 years, which improves GSG nesting habitat, but produces poorer prairie dog habitat. (Figures 7 and 8).

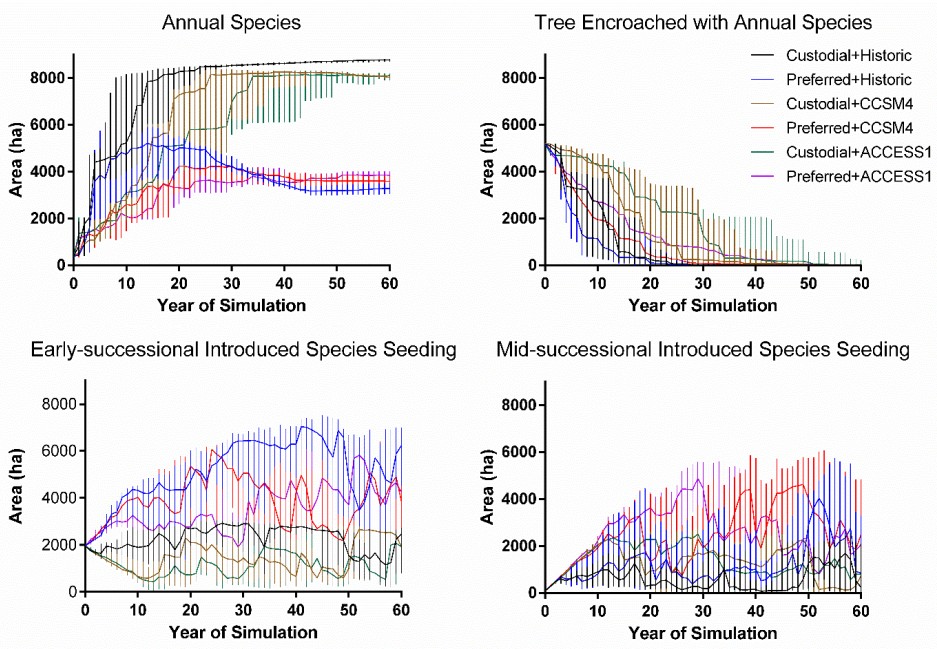

**Figure 7.** Median area (ha) of dominant treated and recipient classes of montane sagebrush steppe in the Black Mountains. Error bars are 25% and 75% quartile range based on 10 Monte Carlo replicates.

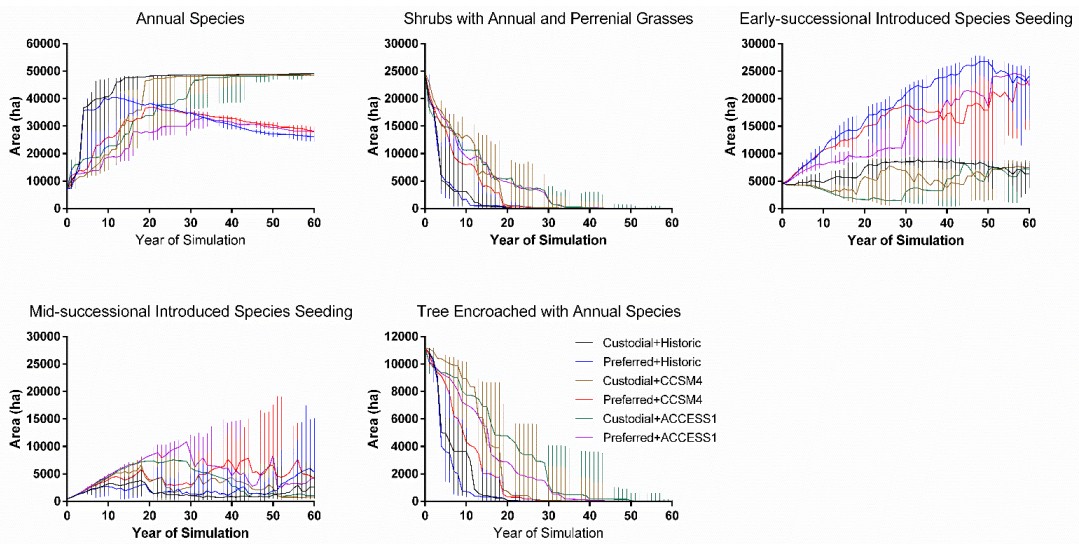

**Figure 8.** Median area (ha) of dominant treated and recipient classes of Wyoming big sagebrush–upland in the Black Mountains. Note the different y-axis scales. Error bars are 25% and 75% quartile range based on 10 Monte Carlo replicates.

Differences in UED and vegetation due to climate differences were primarily driven by large differences in fire regimes. For montane sagebrush steppe and Wyoming big sagebrush–upland, the ACCESS1 LOCA annually burned less than the CCSM4 LOCA, which in turn burned less than the HISTORIC climate (Table 7). Twice as much fire occurred with HISTORIC climate than with the ACCESS1 LOCA.

**Table 7.** Annual area (ha) burned in Montane Sagebrush Steppe and Wyoming Big Sagebrush–Upland under Custodial Management for all three climates. Error is one 95% C.I. N = 10.

| Scenario | Montane Sagebrush Steppe | Wyoming Big Sagebrush–Upland |
|---|---|---|
| CUSTODIAL + ACCESS1 | 1462 ± 388 | 8392 ± 2122 |
| CUSTODIAL + CCSM4 | 1887 ± 398 | 10,518 ± 1976 |
| CUSTODIAL + HISTORIC | 2806 ± 640 | 15,053 ± 1729 |

*3.4. GSG*

Differences in GSG habitat suitability among scenarios were very significant for CUSTODIAL vs. PREFERRED MANAGEMENT for years 30 and 60, but never significant among climate scenarios (Table 8). The lack of climate differences was expected within the PREFERRED MANAGEMENT climate scenarios because the different implementation rates of restoration actions among climates for each management scenario were designed to minimize climate differences.

**Table 8.** Average habitat suitability for GSG in the Black Mountains. Habitat suitability scales from 0% to 100%. Average habitat suitability for year 0 of simulation was 50.5%. Difference between means was analyzed using a $2 \times 3$ factorial analysis of variance (ANOVA; 2 = management levels, 3 = climate levels). Degrees of freedom (df) for management, climate, interaction factors, and error, respectively, were $(2 - 1) = 1$, $(3 - 1) = 2$, $(2 - 1) \times (3 - 1) = 2$, and $2 \times 3 \times (10$ replicates $- 1) = 54$.

| Scenario Name | Year 30 [@] | | Year 60 * | |
|---|---|---|---|---|
| | Mean [#] (n = 10) | ±1 95% C.I. | Mean (n = 10) | ±1 95% C.I. |
| CUSTODIAL + HISTORIC | 43.45 [a,i] | 0.46 | 43.86 [a,i] | 0.44 |
| PREFERRED + HISTORIC | 46.95 [a,i] | 0.33 | 48.73 [b,i] | 0.95 |
| CUSTODIAL + ACCESS1 | 43.87 [a,i] | 0.54 | 43.57 [a,i] | 0.56 |
| PREFERRED + ACCESS1 | 47.56 [b,i] | 0.75 | 47.98 [b,i] | 0.53 |
| CUSTODIAL + CCSM4 | 43.72 [a,i] | 0.64 | 43.79 [a,i] | 0.57 |
| PREFERRED + CCSM4 | 47.24 [b,i] | 0.83 | 48.82 [b,i] | 0.79 |

[#] Habitat suitability were transformed at years 30 and 60 to homogenized variances that were higher for lower average values using the same Box–Cox transformation $(\text{HS\_YR30}^{-4.999997} - 1)/(-4.999997)$. [a] Different letters (a and b were for management effects, whereas i and ii were for climate effects) indicate significant differences based on preplanned contrasts. [@] At year 30, differences were significant (a) at $p < 0.0001$ for Minimum vs. Preferred, (b) but nonsignificant at $p = 0.27$ for climate. MSE for Management = 253.111, Climate = 1.487, Management $\times$ Climate = 0.023, and Error = 1.112. * At year 60, differences were significant (a) at $p < 0.0001$ for Minimum vs. Preferred but nonsignificant at $p = 0.28$ for climate effects. MSE for Management = 381.402, Climate = 1.516, Management $\times$ Climate = 0.132, and Error = 1.182.

Compared to the CUSTODIAL + HISTORIC scenario at year 60, sizable improvements to GSG habitat suitability were achieved with all PREFERRED MANAGEMENT scenarios at year 60 (blue colors in Figure 9). Extensive improvements (blue colors) were primarily found in the north-central and southwestern parts of this landscape. In general, these were areas that were already better habitat without management (Figure 9A). While areas of degradation mostly correspond to nonhabitat and private lands where no money was spent, decreasing habitat suitability in the northeast corner, central-western boundary, and central southern area was observed in all scenarios.

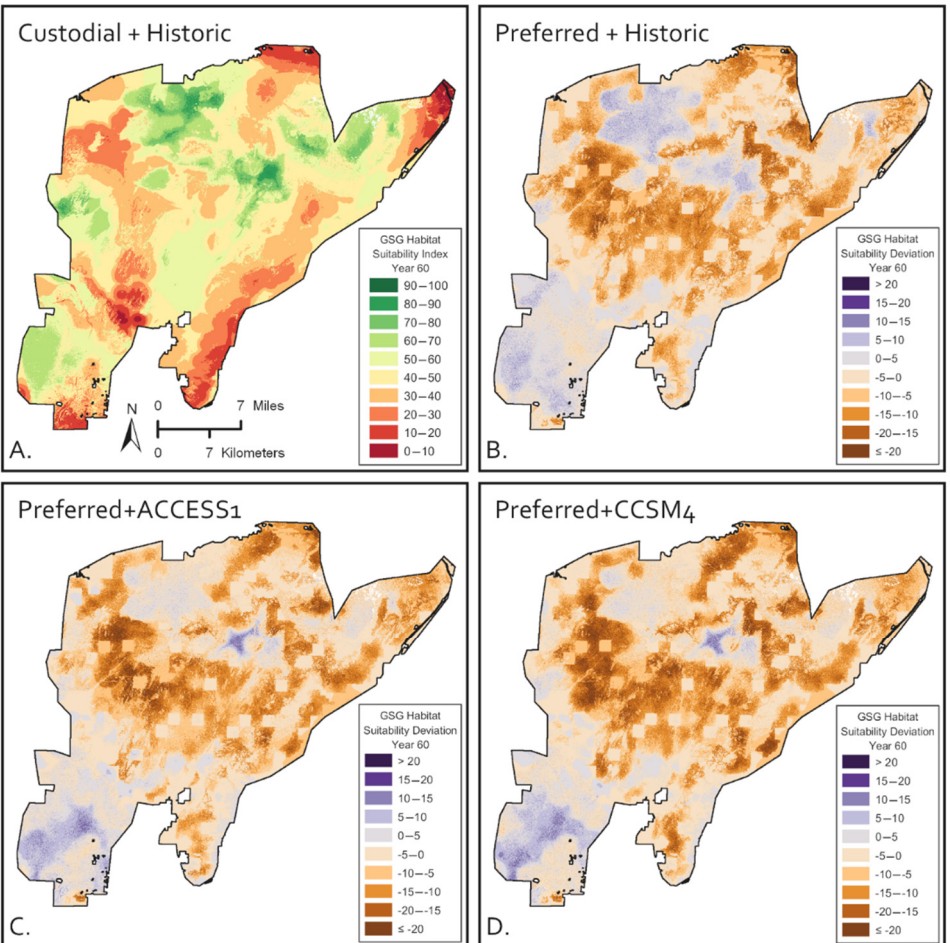

**Figure 9.** Habitat suitability for GSG in the Black Mountains at year 60 for the CUSTODIAL + HISTORIC scenario (**A**), and absolute deviations from that standard for the PREFERRED + HISTORIC (**B**), PREFERRED + ACCCES1 (**C**), and PREFERRED + CCSM4 (**D**) scenarios, based on 2013 five-meter RapidEye satellite imagery.

While all climate scenarios produced statistically similar spatially averaged habitat suitability improvements (i.e., nonspatial estimate; Table 8), maps reveal spatial differences in habitat suitability among all scenarios. The most striking difference is the habitat degradation in the north-central portion of the landscape, which is more pronounced in the PREFERRED + CCSM4 scenario than in the PREFERRED + HISTORIC and PREFERRED + ACCESS1 scenarios (Figure 9). The PREFERRED + ACCESS1 scenario had more (spatially) substantial improvements in GSG habitat compared to others (darkest blue areas).

*3.5. UPD*

Values of average habitat suitability for UPD were consistently smaller by 1.9 to 5.9 in absolute value under CUSTODIAL MANAGEMENT than under PREFERRED MANAGEMENT scenarios at years 30 and 60 (respectively, $p < 0.00001$ and $p < 0.00001$; Table 9). Strong differences existed among climate scenarios at year 30 ($p < 0.00001$) but were not significant at year 60 ($p < 0.79$; Table 9). Habitat suitability was about 5.8% significantly higher under HISTORIC climate than under LOCA-based scenarios for year 30 ($p < 0.00001$). The CCSM4 climate scenario showed about 8.3% greater habitat suitability than under the ACCESS1 climate at year 30 ($p < 0.00001$).

**Table 9.** Average habitat suitability for UPD in the Black Mountains. Habitat suitability scales from 0% to 100%. Average habitat suitability for year 0 of simulation was 48.4%. Difference between means was analyzed using a 2 × 3 factorial analysis of variance (ANOVA; 2 = management levels, 3 = climate levels). Degrees of freedom (df) for management, climate, interaction factors, and error, respectively, were (2 − 1) = 1, (3 − 1) = 2, (2 − 1) × (3 − 1) = 2, and 2 × 3 × (10 replicates − 1) = 54.

| | Year 30 [@] | | Year 60 [*] | |
|---|---|---|---|---|
| **Scenario Name** | **Mean [#] (n = 10)** | **±1 95% C.I.** | **Mean (n = 10)** | **±1 95% C.I.** |
| CUSTODIAL + HISTORIC | 44.31 [a,i] | 0.83 | 44.15 [a,i] | 0.69 |
| PREFERRED + HISTORIC | 46.20 [b,i] | 1.37 | 47.34 [b,i] | 0.79 |
| CUSTODIAL + ACCESS1 | 35.75 [a,ii] | 3.19 | 41.88 [a,i] | 3.10 |
| PREFERRED + ACCESS1 | 39.04 [b,ii] | 2.66 | 45.74 [b,i] | 2.68 |
| CUSTODIAL + CCSM4 | 38.04 [a,ii] | 4.93 | 43.38 [a,i] | 1.40 |
| PREFERRED + CCSM4 | 44.53 [b,i] | 2.00 | 45.99 [b,i] | 1.51 |

[#] Habitat suitability were transformed at years 30 and 60 to homogenized variances that were higher for lower average values, respectively, using Box–Cox transformations $(HS\_YR30^{4.241582} - 1)/(4.241582)$ and $(HS\_YR60^{4.999997} - 1)/(4.999997)$. [a] Different letters (a and b were for management effects, whereas i and ii were for climate effects) indicate significant differences based on preplanned contrasts. [@] At year 30, preplanned contrasts were significant (a) at $p < 0.000862$ for Custodial vs. Preferred, (b) significant at $p = 0.000002$ for Historic vs. ACCESS1 + CCSM4, and (c) significant at $p = 0.000327$ for ACCESS1 vs. CCSM4. MSE for Management = 3.15, Climate = 0.000051, Management × Climate = 0.00009, and Error = 0.000005. [*] At year 60, preplanned contrasts were significant at $p < 0.000002$ for Custodial vs. Preferred, (b) at $p = 0.15$ for the Climate effect, and (c) $p = 0.79$ for Management × Climate. MSE for Management = 0.000085, Climate = 0.000006, Management × Climate = 0.000001, and Error = 0.000003.

PREFERRED MANAGEMENT improvements of UPD habitat suitability were observed in many areas that were scattered throughout the landscape. These maps contrast with the CUSTODIAL + HISTORIC scenario, in which improved habitat suitability occurs in larger, defined areas (Figure 10). In general, spatially averaged habitat suitability was not different among climate scenarios (Table 9). Habitat improvements were more spatially noticeable for the PREFERRED + CCSM4 scenario than for the PREFERRED + ACCESS1 scenario, and the smallest improvements were found in the PREFERRED + HISTORIC scenario by year 60. Several areas of degradation were observed in all PREFERRED MANAGEMENT scenarios, but the most noteworthy area of diminished habitat occurs in the west-central part of the landscape in an area that overlaps with the largest UPD priority management zone (Figures 3 and 10).

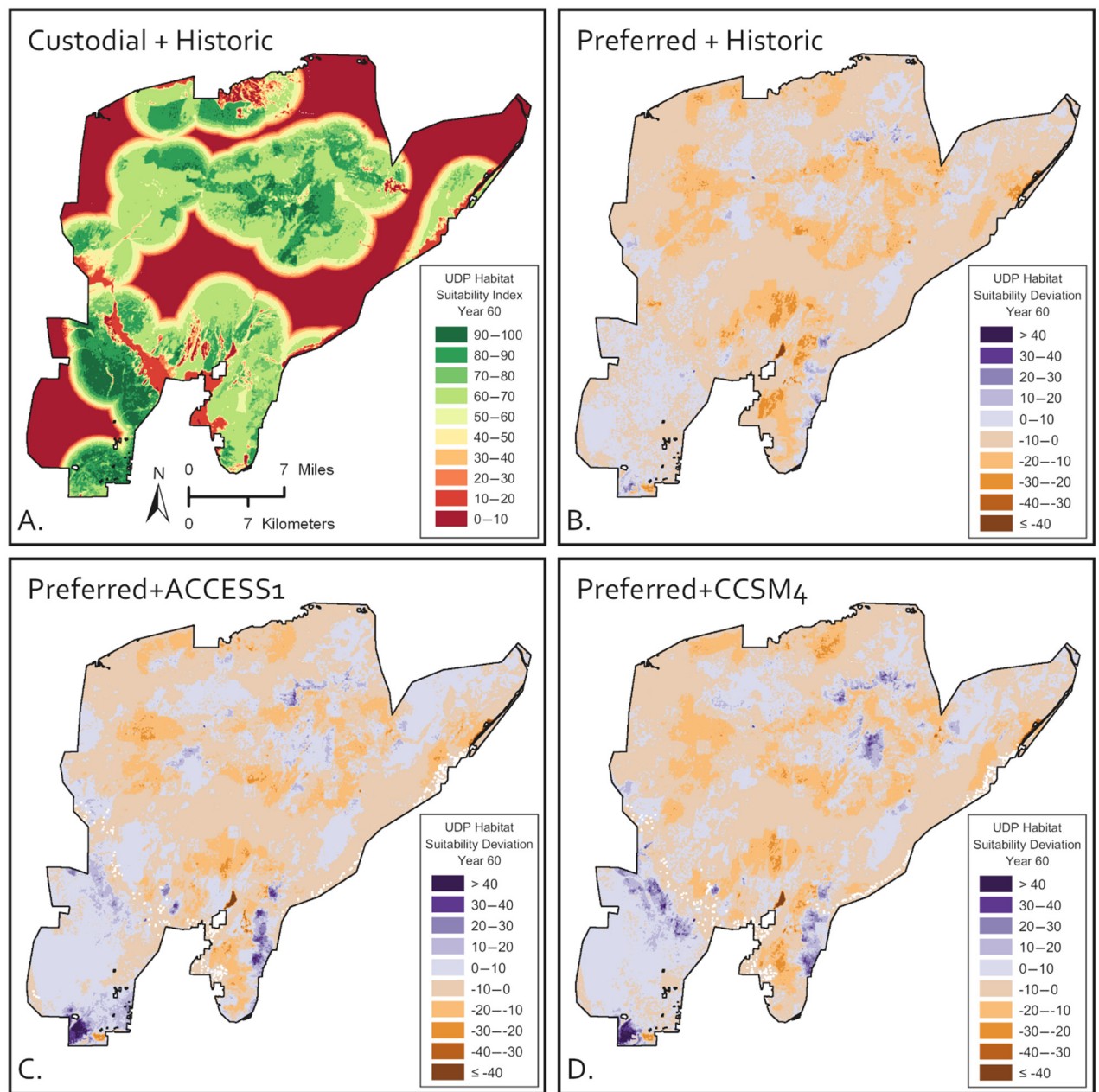

**Figure 10.** Habitat suitability for UPD in the Black Mountains at year 60 for the Custodial + Historic (**A**) scenario, and absolute deviations from that standard for the Preferred + Historic (**B**), Preferred + ACCCES1 (**C**), and Preferred + CCSM4 (**D**) scenarios, based on 2013 five-meter RapidEye satellite imagery.

## 4. Discussion

### 4.1. Unified Ecological Departure vs. Habitat Suitability

UED and habitat suitability indices are different metrics as ecological departure is non-spatial and estimated per ecological system [5], whereas the latter is spatial and estimated for whole landscapes [4]. Managers may use both metrics to justify management actions. However, standard practices of vegetation management invariably focus on prescriptions tailored to each ecological system's soil potential [9,35] that may be better matched to ecological departure assessments. The choice of management objectives, therefore, needs to be explicit about prioritizing actions for ecological systems relative to benefits for special single species such as GSG or UPD.

UED may have limited value to measure ecological improvement for sagebrush-dominated systems at lower elevations where GSG and UPD spend a considerable portion

of their life cycles [14,15,33]. High UED values might encourage some managers to treat vegetation classes that contribute most to departure with the expectation that UED will decrease over time in treated areas (e.g., Figures 7 and 8). This expectation is not met at lower elevations because restoration creates uncharacteristic vegetation classes where annual grasslands and tree-encroached shrublands are replaced with mixed introduced and native species seedings. These large seedings contributed to departure as any other uncharacteristic classes [5,11]. Higher-elevation systems and lower-elevation moist systems will be more likely to show UED reductions with treatments because they already contain reference classes, and pure native species mixtures can be seeded at higher elevations [35]. Native species seedings also become undistinguishable from reference classes over time.

While UED was not consistently responsive to large amounts of restoration, habitat suitability indices responded to the effects of treatments and climate (Tables 8 and 9, Figures 9 and 10). Despite million-dollar investments in restoration, significant differences in habitat suitability indices between CUSTODIAL and PREFERRED management rarely exceeded 5% in absolute value for GSG (Table 8) and 10% for UPD (Table 9). This was expected because *RSF*s incorporate important physical characteristics of the landscape that do not change at all, such as distance to roads, leks, and moist summer vegetation for chick rearing (Supplemental File S6) [33,34]. Therefore, absolute values of 5% and 10%, respectively, change in habitat suitability might represent a sizable improvement to GSG and UPD demography. The success of habitat suitability might also mirror reduction in UED (smaller values are better) for multiple systems if the focal species requires a diversity of habitat resources, such as GSG [14,33].

### 4.2. Climate Differences

Climate differences accounted for large variations in the areas of burned tree-encroached shrublands (Table 7), annual grassland, seedings, and seedings that failed due to drought during their establishment (Figures 7 and 8). Double the area burned with HISTORIC climate than the ACCESS1 LOCA, while the CCSM4 LOCA showed intermediate values (Table 7). Progressively drier climates with longer drought periods and less fire decreased UED but incurred a greater management cost by requiring additional tree removal and repeat seedings up to the allowed annual budget ceiling. One goal of the project method was to compensate for climate differences among scenarios by increasing restoration levels.

BLM staff were keenly interested in the effects of different future climates on the success of native and introduced species seedings. Since 2015, the BLM Cedar City Field Office treated about 17,915 ha of rangeland in the Black Mountains, of which about 6296 ha were seeded with the expectation of maintaining comparable high levels of seeding in the future. Using the SPI drought measure, BLM seeding experts proposed that 24 months of drought evaluated in years $t - 1$ and $t - 2$ (t is the current year) and 36 months of drought evaluated over years $t$, $t + 1$, and $t + 2$ increased the likelihood of seeding failure. The CCSM4 LOCA with greater spring and early summer rains would show less seeding failure than the ACCESS1 LOCA, which is drier during all seasons. Results indicate that seedings will fail more under the drier ACCESS1 climate, but drier climate will also favor later successional seedings because of reduced fire activity. If the ACCESS1 LOCA is the likely future then front-loading extensive restoration that includes seedings in the next decade is more important for management success than spreading restoration into the future (Figures 7 and 8). If the CCSM4 LOCA is believed to be more likely then managers have about 30 years to spread seedings that will benefit from increased spring and early summer precipitation favorable to seedling survival (Figures 7 and 8). Eventually, higher temperatures and evapotranspiration are expected to overcome the benefits of more spring moisture. The cost of drier climate observed in the ACCESS1 LOCA compared to historic climate to managers was about USD 2 million more during the first 30 years of intense management, but about USD 2.4 million less from years 31 to 60 (Table 6). The bulk of extra expenditures during the first 30 years was attributable to tree removal to make up for the

lack of stand-replacing fire under drier conditions. Greater fire activity was modeled to be more likely if fine fuels first accumulate two years prior to the current year due to above average precipitation, followed by the focal year with dry fuels [36,37]. Therefore, drier climate before a current dry year resulted in substantially less fire. After year 30, less fire resulted in less annual grasslands created, which spares managers the cost of additional non-native annual grassland treatments.

*4.3. GSG vs. UPD*

UPD is a listed species [15], whereas GSG is not. In addition, the Black Mountains area is the source population for regional UPD reintroductions to other promising colony sites. GSG was, however, used to locate most treatments because the narrow habitat requirements of UPDs can be handled as special restoration projects with dedicated treatments not shared in GSG management zones. GSG habitat requirements are broad and poorly align with UPD needs in degraded landscapes. Restoration with a focus on GSG achieves broad conservation goals benefiting other shrubland-dependent and wet meadow species [38].

Managing for GSG alone, however, has some drawbacks because what is good for GSG may not be good for range management. For example, degraded forms of standing sagebrush have nesting value to GSG, whereas mowing and then seeding native and/or introduced herbaceous species and planting plugs of sagebrush to improve range condition renders nesting and brood-rearing value nil or negative until sagebrush is abundant enough to provide nesting habitat [33,34]. Whether or not managers restore degraded sagebrush depends on local agency directives and a short-term versus long-term view of habitat quality and resilience to fire.

**5. Conclusions**

(a). It is feasible to rapidly build expert-driven habitat suitability indices for planning if statewide data and expertise exist during the time that local data are collected. While some statewide demographic data existed for northern Utah populations of GSG in 2013, no or only minimal bird movement data had been collected from the southwestern populations of Utah where vegetation receives less precipitation and experiences greater temperature compared to northern populations, and lies only 100 km from the Mojave Desert. GSG experts were not convinced that *RSF*s from the north applied to the Black Mountains, but collecting more demographic data would require significant funding and a decade of field work in southwest Utah to achieve significant statistical power to define *RSF*s. Therefore, experts were willing to propose the strongest *RSF*s in a three-hour workshop that applied to all populations of GSG, which were later revised using statistically defensible demographic data from southern populations in central Nevada [2,33,34].

(b). Extensive restoration is predicted to accomplish management goals for ecological systems and for GSG regardless of future climate. Active management scenarios for any future climate improved UED, especially at higher elevations, and GSG habitat suitability compared to CUSTODIAL MANAGEMENT. The levels of implementation of proposed actions were based on a track record of past funding by BLM staff willing to focus actions in the next 10–15 years before climate is expected to become hotter and potentially drier; therefore, expectation of success is realistic. In addition, extensive restoration is likely to reduce fire activity largely driven by non-native annual fuels compared to piecemeal seedings incapable of stopping or slowing large uncharacteristic fires.

(c). There is an incentive to front-load restoration of degraded subxeric ecological systems during the next 10 to 30 years if climate is expected to be drier, especially if spring and early summer precipitation is predicted to decrease. The success of seedings, which are commonly used in most widespread restoration, decreases with drought prior and after the year of seeding. All climate scenarios predict hotter temperatures that will eventually cause greater evapotranspiration, regardless that greater spring and sum-

mer precipitation is predicted for the CCSM4 LOCA, whereas the ACCESS LOCA will be drier in all seasons. The more severe effects of these scenarios, especially the magnitude of future evapotranspiration, are expected to manifest themselves after 30 years in the future when different global circulation models start differentiating [27].

(d). This study uniquely spans fields of spatial STSM [8,10], management of western USA rangelands [1,35,39], at-risk species habitat suitability modeling [4], and incorporation of climate change scenario effects on ecological systems supporting at-risk species [10]. The Nature Conservancy is a science-based conservation organization that extensively uses existing data but does not generally generate new science. For this project, we innovatively brought these four fields together to answer pressing management needs of the BLM. While a few studies had pairwise combined STSM with forest management [8], range management [6,10], or climate scenarios [10,39], but not species assessments, it is only recently that the ST-Sim software has allowed simultaneous integration of spatial and temporal components to achieve innovative modeling [8].

**Supplementary Materials:** The following are available online at https://www.mdpi.com/article/10.3390/cli9050079/s1, Table S1: Description of Ecological Systems and Vegetation Classes for the Black Mountains UT, File S2: Custom Python computer program to conduct resampling, Table S3.1: Rasters uploaded to ST-Sim, File S4: Creating transition multipliers for ST-Sim, File S5: Range Shifts, File S6. Greater sage-grouse RSFs and habitat suitability.

**Author Contributions:** Conceptualization, L.P., K.B., and J.T.; methodology, L.P. and K.B.; software, K.B. and L.P.; formal analysis, L.P., K.B., T.A., and S.B.; investigation, L.P., K.B., T.A., and J.T.; resources, K.B., T.A., S.B., and L.P.; software, K.B. and S.B.; data curation, T.A. and S.B.; writing—original draft preparation, L.P.; writing—review and editing, L.P., K.B., J.T., T.A., S.B., E.Y., and D.F.; visualization, S.B., K.B., L.P., and J.T.; supervision, L.P. and E.Y.; project administration, E.Y.; funding acquisition, D.F., E.Y., L.P., and J.T. All authors have read and agreed to the published version of the manuscript.

**Funding:** Utah's Department of Interior Bureau of Land Management award L16AC00162 to E.Y. at The Nature Conservancy in Utah funded the project.

**Institutional Review Board Statement:** Not applicable.

**Informed Consent Statement:** Not applicable.

**Data Availability Statement:** All ST-Sim model data can be download without password from tnc.box.com @: https://tnc.box.com/s/9mwyghq0962mk4ybtdohel12tua5ih05, accessed on 21 March 2020. Authors currently use version v.2.2.27 of Syncrosim (ST-Sim module is part of Syncrosim), whereas the original model was completed in an older version. V2.2.27 will open and upgrade older model databases or a copy of the older software can be obtained from www.apexrms.com (accessed on 23 April 2021).

**Acknowledgments:** Spatial Solutions Inc. conducted remote sensing with The Nature Conservancy's field support. The following contributed to ST-Sim models and habitat suitability indices: Gary Bezzant, Rhett Boswell, Nathan Brown, Becky Bonebrake, Paul Briggs, Elizabeth Burghard, Renee Chi, Dave Dalhgren, Nicki Frey, Jimi Gragg, Betsy Herrmann, Jay Martini, Jason Nicholes, Russ Norvell, Doug Page, Christine Pontarolo, Jason Robinson, and Danny Summers.

**Conflicts of Interest:** The authors declare that they have no conflicts of interest. In 2017, The Nature Conservancy's Utah Field Office awarded to the Bureau of Land Management Cedar City Field Office the Utah Conservation Partner Award. The funders had no role in the design of the study; in the collection, analyses, or interpretation of data; in the writing of the manuscript, or in the decision to publish the results.

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
