# Peer review of "Landscape Conservation Forecasting for Data-Poor at-Risk Species on Western Public Lands, United States"

_climate, doi:10.3390/cli9050079_

Round 1

Reviewer 1 Report

I'm sure that other reviewers will also mention this - but lines 1-39 and the first half of line 40 should be deleted.  

Line 150 - what is "spatial data of post-fire seeding treatments"?  At first I thought this was referring to the field data referenced in the previous sentence, but I am thinking that this is the spatial location of post-fire seeding treatments?  If so, I suggest using "spatial location" instead of "spatial data".  If not, please clarify what Line 150 is referring to.  

Line 184 - was this really suppression?  Many ecological systems in the western US had active management by indigenous people for thousands of years prior to European settlement.  Did these ecosystems also have indigenous management with fire?  If so, fire exclusion (instead of suppression) may be the more appropriate terminology to use.  

Table 4 is difficult to read - bolding the "Ecological Systems" would help distinguish them from the "actions".  Left rectifying the ecological systems might also help.  

Also in Table 4 - what is "plateau" in the list of actions?  I did a word search in the document but could not find a definition.  

Since the data collection and analysis occurred 8-9 years ago,  it would be a good addition to the paper (or perhaps a subsequent paper) to look at how the vegetation has changed in the intervening period with or without restoration.  

Lines 468-469 mention organisms other than sage grouse and prairie dogs that may need special attention.  While weeds are referred to earlier in the document, mule deer and migratory birds are not.  Since this is in the results section, information about mule deer, migratory birds and weed management should be included in the methods as well.  Same for pine nut harvesting (line 478).  

Excellent discussion of the ecological departure vs. habitat suitability in sagebrush ecosystems.  

Author Response

  1. Line 8: Deleted Las Vegas as the city is Moab (our error), not Las Vegas, which is anyway in NV, not UT.
  2. Delete lines 1-31, and first half of lines 40:
    Response: Surely this is wrong as this means to delete the title, authors’ names and affiliations, and abstract. However, the text from Lines 31 to half of line 40 was added by the editorial office but not the authors and was deleted.
  3. Line 150: Rewrite sentence “In addition to these observations, spatial data of post-fire seeding treatments were used to improve the interpretation of imagery.”
    Response: The sentence was changed to” In addition to these observations, spatial data of seeding treatments that followed fires but predated our mapping were used to improve the interpretation of imagery.”
  4. Line 184: Replace “fire suppression” with “fire exclusion.” Also was there indigenous burning?
    Response: (a) We agree. The word “suppression” was replaced with “exclusion.” (b) Also, there are no data on indigenous burning in these systems (unlike others in geographies receiving more rain), except for strong inference for burning aspen-mixed conifer communities, which are not in the Black Mountains. Aspen woodlands, which have no to very limited potential to support conifer species, might have been burned but the evidence is lacking as conifer fuel is needed to carry fire in aspen woodlands. Because aspen and pinyon-juniper woodland ecologists and anthropologists, which helped us fine-tune our original models in 2010, strongly suspected use of fire in aspen woodlands and in gaps between pinyon trees for tobacco production and pinecone roasting, we added a minor component of indigenous fire to these systems. For other systems, we suspect some indigenous burning, but we have no hard evidence nor inference.
  5. Table 4: Bolding and left-justifying ecological systems to increase readability.
    Response: Done. In the submitted MS, ecological systems were left-justified but not bolded. The editorial office changed the formatting.
  6. Table 4: Define Plateau.
    Response: I added a definition in the legend to Table 4’s caption. Plateau® is the commercial name for the herbicide imazapic that prevents germination of annual plant species. Plateau is so commonly used in the western USA that we forget that some audiences might not know.
  7. Because study was done 8-9 years ago, could authors report on vegetation management accomplished since in this paper or maybe a future paper.
    Response: We wish we could follow up. Various seedings and other range management actions have been implemented as a result of this modeling but The Nature Conservancy would only conduct a follow-up analysis if funding could be found to do so, which has not been the case. We have discussed a proposal for remote sensing change detection, but it is hard for local offices of government agencies to obtain such “follow-up” funding when on-the-ground restoration needs are so great. We did not add anything to the manuscript.
  8. Lines 468-469: Add to methods text about mule deer, migratory birds, and weed management to methods since these are in results.
    Response: We will not add to methods because mule deer and migratory birds were never studied here. Weed management methods was already described in the treatments that include Plateau and weed control in wet meadows and riparian systems. The inclusion of mule deer and migratory birds as a criteria of system inclusion was a request from BLM. Therefore, the sentence “The Custodial Management scenario identified ecological systems that will further degrade, stay very highly degraded, or may need special attention for GSG, UPD, mule deer (Odocoileus hemionus), migratory birds, or weed management.” was changed to two sentences: “The Custodial Management scenario identified ecological systems that will further degrade, stay very highly degraded, or may need special attention for GSG, UPD, or weed (non-native annual species and exotic riparian forbs and trees) management. In addition, BLM staff requested that we include systems that are important to habitat of mule deer (Odocoileus hemionus) and migratory birds, although those species were not assessed here.”

Reviewer 2 Report

The article shows spatial model simulations of state and transition of ecological systems for different scenarios up to 60 years, while coupled with expert-developed habitat suitability indices. The authors conclude that the driest climate most affected prairie dog ecological departure and habitat suitability only at 30 years and that different climates influenced spatial patterns of prairie bear habitat suitability, but non-spatial values did not change.
The article is rigorous and very well structured. I believe that the authors need to make some modifications to make the article publishable. Below are some suggestions that should be taken into consideration by the authors.
1. The authors explain many results based on climatic impacts on fire, vegetation succession and restoration, but it would be interesting to have more specific details because it is expected that the restoration of the front load will benefit under a drier future climate.
2. Justify in the conclusions, more practically and with examples if possible, why it is feasible to develop expert-based habitat suitability indices for planning if statewide data and knowledge exist during the time of local data collection; 
3. Detail why extensive restoration is expected to meet ecological systems and GSG management objectives regardless of future climate. 
4. Further detail in the conclusions what the incentive is to advance restoration of degraded subxeric ecological systems over the next 10-30 years if the climate is expected to be drier, especially if spring and early summer precipitation is expected to decrease.
5. Increase impactful literature references in discussions of results.
6. The state of the art can also be enriched with more impact literature references.

Author Response

  1. Explain with more details the effects of climate on seedings and restoration.
    Response: The details requested by the reviewers are, in part, in the supplemental material, but we inserted key climate effects in the manuscript. We inserted key climate details in lines 680-683 (original MS) for fire effects and among lines 666-668 (original MS) where this applies to seedings. (a) The sentences ”Since 2015, the BLM Cedar City Field Office treated about 17,915 ha of rangeland in the Black Mountains, of which about 6,296 ha were seeded with the expectation of maintaining comparable high levels of seeding in the future. Using the SPI drought measure, BLM seeding experts proposed that 24 months of drought evaluated in years t-1 and t-2 (t is the current year) and 36 months of drought evaluated over years t, t+1, and t+2 increased the likelihood of seeding failure. The CCSM4 LOCA with greater spring and early-summer rains would show less seeding failure than the ACCESS1 LOCA, which is drier during all seasons.” after the sentences “BLM staff were keenly interested in the effects of different future climates on the success of native and introduced species seedings.” Note too that the sentence starting with “In 2016, the BLM Cedar City Field Office seeded about 6,478 ha…” was modified as BLM has completed a large amount of restoration after our project (see above).  (b) We added the sentence “Greater fire activity was modeled to be more likely if fine fuels first accumulate two years prior to the current year due to above average precipitation, followed by the focal year with dry fuels [34-35]. Therefore, drier climate before a current dry year resulted in substantially less fire.” after the sentence “The bulk of extra expenditures during the first 30 years was attributable to tree removal to make up for the lack of stand-replacing fire under drier conditions.” of lines 680-682.
  2. Add examples and more practical information to conclusion #1.
    Response: We agree as this would be more interesting and important (this point actually started this whole process!), but the conclusion is now longer. Conclusion #1 was rewritten from one simple sentence to the following paragraph: “ It is feasible to rapidly build expert-driven habitat suitability indices for planning if state-wide data and expertise exist during the time that local data are collected. While some statewide demographic data existed for northern Utah populations of GSG in 2013, no or only minimal bird movement data had been collected from the south-western populations of Utah where vegetation receives less precipitation and experiences greater temperature only 100 km from the Mojave Desert compared to northern populations. GSG experts were not convinced that RSFs from the north applied to the Black Mountains but collecting more demographic data would require significant funding and a decade of field work in southwest Utah to achieve significant statistical power to define RSFs. Therefore, experts were willing to propose the strongest RSFs in a three-hour workshop that applied to all populations of GSG, which were later revised using statistically defensible demographic data from southern populations in central Nevada [2, 31-32];”
  3. Explain why extensive restoration is expected to lead to success regardless of future climate.
    Response: We added the following sentences after the original single-sentenced second conclusions: “Active management scenarios for any future climate improved UED, especially at higher elevations, and GSG habitat suitability compared to Custodial Management. The levels of implementation of proposed actions were based on a track record of past funding by BLM staff willing to focus actions in the next 10-15 years before climate is expected to become hotter and potentially drier; therefore, expectation of success is realistic. Also, extensive restoration is likely to reduce fire activity largely driven non-native annual fuels compared to piecemeal seedings incapable of stopping or slowing large fires;”
  4. Explain the 10-30 years period to front-load restoration.
    Response: To explain the 10-30 years grace period for restoration, we added the predictions from IPCC in the context of our work after the original single sentence of the last conclusion: “The success of seedings, which are commonly used in most widespread restoration, decreases with drought prior and after the year of seeding. All climate scenarios predict hotter temperatures that will eventually cause greater evapotranspiration, regardless that greater spring and summer precipitation is predicted for the CCSM4 LOCA whereas the ACCESS LOCA will be drier in all seasons. The more severe effects of these scenarios, especially the magnitude of future evapotranspiration, are expected to manifest themselves after 30 years in the future when different Global Circulation Models start differentiating [25].”
  5. Add high impacts literature references to discussion of results.
    Response: We added references to the discussion using existing references and added three new references to the discussion and literature cited  (Taylor and Beaty, Westerling and Bryant, Hanser and Knick). One major challenge (actually, inability) of this work was to find literature that encompassed even two of the four major areas presented in this manuscript (climate scenarios and their effects on vegetation through altered ecological processes, at-risk species habitat suitability modeling, western rangeland management, and state-and-transition simulation modeling of western landscapes). The power of our work is to bring all four together; therefore, we used the most appropriate references applicable to the under-studied Great Basin realizing these are not always the most impactful as measured by journal impact factors.
  1. Enrich state-of-the-art with more impactful literature references.
    Response: See response above. Also, we added a fourth (d) conclusion about state-of-the-art science and one new high-impact reference, we hope. We are not sure where in the MS and what the reviewer expected. We are also not too enthusiastic about this fourth Conclusions as it is borderline prideful.

Reviewer 3 Report

General comments
This paper describes "Landscape conservation forecasting for data-poor at-risk species on western public lands". The manuscript is well-structured and within the scope of the Journal. I found that it is beneficial for scientists and researchers in terms of information presented. I will support this article after a revision by incorporating the suggested correction.

Minor comments:
Comment 1: Abstract should have a statement about the research gap and focus of the current study. 
Comment 2: Introduction needs to be revised a bit, state-of-the-art-of the study is missing (kindly define the risk to climate;  Estimation of risk to the eco-environment and human health of using heavy metal in the Uttarakhand Himalaya, India. Applied Sciences 10 (10.3390/app10207078), 7080ï¼›Climate vulnerability index-measure of climate change vulnerability to communities: a case of rural Lower Himalaya, Indiaï¼›R Pandey, SK Jha Mitigation and Adaptation Strategies for Global Change 17 (5), 487-506, 2012

Comment 3: English language needs to revise thoroughly throughout the manuscript to avoid minor errors. At some places, spacing between words is missing
Comment 4: Multiple references are of no use for a reader and can substitute even a kind of plagiarism, as sometimes authors are using them without proper studies of all references used. In this case each reference should be justified by it is used and at least short assessment provided.
Comment 5: Resolution of figures 1 needs to be improved for better visibility to readers, and further give the latitude and longitude
Comment 6: Discussion section needs to be strengthened and comparatively supported with few more related reported studies available in the literature.

Comment 7: Conclusion should be shortened and reasonable.
Technical Comments

Comment 8: Explain the implications and how this study will be helpful for the management of natural resources in the target area.

Author Response

  1. Add statements to the abstract something about the focal point and research gaps.
    Response: (a) Focal point: The focus of the paper is already in the abstract in the first two sentences “Managing vast federal public lands governed by multiple land use policies creates challenges when demographic data on at-risk species are lacking. The U.S. Bureau of Land Management Cedar City Field Office used this project in the Black Mountains (Utah) to inform vegetation management supporting at-risk greater sage-grouse and Utah prairie dog planning.” Therefore, we will not add to the abstract.  (b) We do not think it is appropriate to add text about research gaps to a 200-word maximum abstract where we are at 201 words. Many obscure academic research gaps exist that these authors will not investigate and covering them might fill another paper that should be written by other people.
  2. Add to the state-of-the-art in the Introduction and citations from Indian climate science papers mostly focused on climate change effects on human impacts and heavy metals.
    Response: (a) The reviewer is requesting that we cite two manuscripts, presumably ones they published. The two manuscripts’ contents based on geographies from India are not at all related to this study. There are 10,000s of publications on climate change and we cannot be expected to review them simply because they address an aspect of climate change and if they have no relationship to the study. Moreover, one of the manuscript proposed an index of climate vulnerability that we should cite. In the USA, NatureServe proposed such a metric  (https://www.natureserve.org/conservation-tools/climate-change-vulnerability-index) about 10 years ago and we did not discuss it in this paper because it does not inform our study, especially since our habitat suitability models that are fine-grained, quantitative, and dynamic are more powerful. (b) Our Introduction does cover the state-of-the-art of the four fields in this study (Reviewer #2 also asked for this and we added to Conclusions (see(d)). In the Introduction, the state-of-the-art is covered in the sentences: 1st parag on western land management – “Managers must assign high priority to the management of ecological systems supporting at-risk species while considering other natural resource and economic values [1]. Furthermore, managers have limited funds to achieve all management goals.”; 2nd parag on at-risk species habitat suitability modeling – “Local demographic data help guide management actions to improve at-risk species habitat but it is uncommon to have complete demographic data on at-risk species for more than one drought cycle (e.g. 7-year El Niño cycle) reflecting highs and lows of reproductive effort in most western states ─ it is even uncommon for the simplest demographic data [2]”; 3rd parag on state-and-transition simulation modeling (STSM) – “Advances in scenario-based state-and-transition simulation models (STSM) methods and software feasibly allow land managers working with modelers to address these questions that straddle ecological system management planning and single species management [5-9], but seldomly incorporating climate change scenarios [11,37].”; 3rd parag on innovatively blending metrics from STSM and wildlife management –  “Ecological departure can guide site planning management because departure can be partitioned to different vegetation classes [5,9]. However, vegetation responses to management do not always translate to change in species’ habitat suitability because ecological departure is non-spatially estimated by system, whereas habitat suitability is spatially estimated across entire landscapes. Using habitat suitability as a metric of condition to guide vegetation treatment implementation would increase the likelihood of success for the targeted species.”  The one aspect of the state-of-the-art that is not well represented is climate change (it is covered in methods and Supplemental Material). We added the statement that dynamic effects of climate change scenarios on vegetation or focal species are rarely modeled in the third paragraph of the introduction (also above): “…, but seldomly incorporating climate change scenarios [11,37].” Similarly, we added the Landscape Conservation Forecasting can incorporate climate scenarios (in italics), although this is not state-of-the-art (we have been doing this since 2008): “The Nature Conservancy’s Landscape Conservation Forecasting (LCF) method combines vegetation layers obtained from remote sensing with STSMs to compare the effects of alternative management or climate scenarios on vegetation condition and other metrics [5,9,11].” Finally in the last paragraph of the Introduction we added the climate change connection to objectives (second sentence in italics): ”The main impetus of this project was to incorporate habitat suitability as an additional metric of condition into Landscape Conservation Forecasting for two at-risk species managed by the Bureau of Land Management (BLM) in southwestern Utah: Great sage-grouse (Centrocercus urophasianus; hereafter, GSG) and the federally threatened Utah prairie dog (Cynomys parvidens; hereafter, UPD). Both species lacked local, long-term research to inform traditional demography in this region. BLM was concerned about two future climate change risks: (a) Climate warming would affect the habitat of GSG and increase the likelihood of extirpation because this is the most southern mid-elevation population of the species and (b) future 2-year and 3-year droughts, respectively, preceding and following the year of restoration actions including seeding would cause seedings to fail and become dominated by non-native annual species.“
  3. Revise minor errors in text, including missing spaces between words.
    Response: We checked and didn’t find any. However, the editorial office cut-and-paste our original manuscript into another template, which caused some of these problems which we corrected.
  4. Incorrect use of multiple references.
    Response: It’s obvious we had different academic training. We were corrected by past major professors for not using enough references to support our research; therefore, we disagree with the reviewer on this point. One part of the Introduction where we provided context to the use of multiple references is the first parag of Methods (see italics): “Our methods were comprised of three parts: remote sensing of vegetation data, state-and-transition simulation (STSM) modeling, and metric development. We will provide a summary as our methods have evolved over time and been detailed elsewhere [5-6,9,11]. STSM reviews and examples can be found in Czembor and Vesk [7] and Rumpff et al. [12] for uncertainty analysis in older non-spatial STSM, Provencher et al. [11] for application to management, uncertainty accounting, and climate change, and Daniel et al. [8] for theoretical and software advancements in non-spatial and spatial STSM.” With the expectation above, we used at most and uncommonly three references in the rest of the text.
  5. Increase resolution of letters in Figure 1 and add lat/long to figure caption.
    Response: Figure letter resolution increased. Also, we added the coordinates after the original figure caption: “Project area approximate coordinates are 38o01’52.17” N, 113o02’49.87” W.”
  6. Strengthened discussion by using more reported studies.
    Response: We have done this in response to Reviewer #2’s comments.  We added references to support statement for which there were no references and added three new references on climate change and one on species co-benefits.
  7. Conclusions should be shortened.
    Response: They were very short. However, reviewer #2 requested that we add examples and statement to them, which we did, thus making Conclusions longer. We even added a fourth conclusion. The Editor should decide if the shorter (original MS) or longer versions (this revised MS) should be retained.
  8. Explain management implications to target area.
    Response: This is already done in term of front-loading management and even use of appropriate metrics for managers to use for management. We even added to Conclusions to explain this as requested reviewer #2; therefore, we will not add anymore text.

Round 2

Reviewer 2 Report

The authors have attended and responded correctly to all the suggestions made to them, therefore I recommend the publication of the article.

Author Response

Reviewer 2 had no additional comments!